



# The influence of layering and barometric pumping on firn air transport in a 2D model

Benjamin Birner[1], Christo Buizert[2], Till J.W. Wagner[1] and Jeffrey P. Severinghaus[1]

[1]Scripps Institution of Oceanography, University of California San Diego, La Jolla, CA 92093, USA
[2]College of Earth, Ocean and Atmospheric Sciences, Oregon State University, Corvallis, OR 97331, USA

*Correspondence to*: Benjamin Birner (bbirner@ucsd.edu)

**Abstract.** Ancient air trapped in ice core bubbles has been paramount to developing our understanding of past climate and atmospheric composition. Before air bubbles become isolated in ice, the atmospheric signal is altered in the firn column by transport processes such as advection and diffusion. However, the influence of impermeable layers and barometric pumping

(driven by surface pressure variability) on firn air transport is not well understood and cannot be captured in conventional 1-dimensional firn air models. Here we present a 2-dimensional (2D) trace gas advection-diffusion-dispersion model that accounts for discontinuous horizontal layers of reduced permeability. We find that layering and barometric pumping individually yield too small a reduction in gravitational settling to match observations. In contrast, a combination of both effects more strongly supresses gravitational fractionation. Layering locally focuses airflows in the 2D model and thus

amplifies the dispersive mixing resulting from barometric pumping. Hence, the representation of both factors is needed to obtain a more natural emergence of the lock-in zone. Moreover, we find that barometric pumping in the layered 2D model does not substantially change the differential kinetic fractionation of fast and slow diffusing trace gases, which is observed in nature. This suggests that further subgrid-scale physics may be missing in the current generation of firn air transport models. However, we find robust scaling relationships between kinetic isotope fractionation of different noble gas isotope and

elemental ratios. These relationships may be used to correct for kinetic fractionation in future high precision ice core studies.



## 1    Introduction

In the upper 50-130 m of unconsolidated snow above an ice sheet, known as the firn layer, atmospheric gases become gradually entrapped in secluded pores and are eventually preserved as bubbles in the ice below. Antarctic ice core records containing these trapped gases have been critical in informing our understanding of the interplay of past climate and atmospheric trace gas variability over the past 800,000 years (Lüthi et al., 2008; Petit et al., 1999). As atmospheric gases migrate through the firn, they are modified in composition by several competing physical processes which may alter elemental concentrations and isotopic signatures (Buizert et al., 2012; Kawamura

> Box 1| Porous media terminology
>
> *Porosity:* the fraction of (firn) volume filled by gas
>
> *Permeability:* the degree to which a porous medium permits flow to pass through
>
> *Fickian diffusion:* molecular diffusion that is proportional to the concentration gradient as described by Fick's first law
>
> *Tortuosity:* measure of the twistedness of pathways through a porous medium

et al., 2013; Mitchell et al., 2015; Schwander et al., 1988, 1993; Trudinger et al., 1997). Therefore, appropriate corrections must be applied to firn and ice core records to accurately reconstruct atmospheric trace gas histories.

Abundant evidence shows from field observations, high resolution firn density measurements, and comparisons of summer and winter ice, that (near-) impermeable horizontal layers exist in polar firn (Hörhold et al., 2012; Mitchell et al., 2015; Orsi et al., 2015) (Fig. 1). Their significance for firn gas transport remains unclear and motivates this work. Readers who are familiar with the structure of firn and its air transport processes may wish to skip ahead to the last paragraph of this section. To build some intuition about firn transport processes, a simple analytical model of firn air transport is provided in Appendix A.

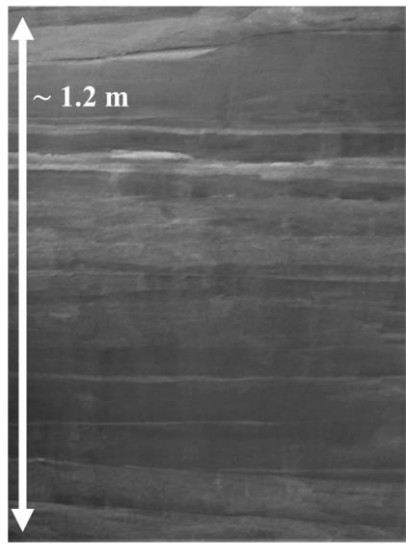

**Figure 1.** Layering of firn photographed in a surface pit at WAIS Divide. Image courtesy of Anaïs Orsi.

We distinguish four main processes affecting the composition of air in firn: diffusion, advection, dispersion and convective mixing. Fickian diffusion, driven by concentration gradients in the firn, is the primary mode of horizontal and vertical transport.




Molecular diffusion also enables gravitational fractionation, or "settling", of trace gases in proportion to their masses (Schwander, 1989; Schwander et al., 1993; Sowers et al., 1989; Trudinger et al., 1997). Gravitational settling leads to an enrichment of heavy isotopes with depth that is described in equilibrium by the barometric equation (Schwander, 1989; Sowers et al., 1989; Craig et al., 1988):

$$\delta_{grav} = \left[\exp\left(\frac{g\Delta m}{R\,T}z\right) - 1\right] \tag{1}$$

where $\delta \equiv \dfrac{r_{sample}}{r_{standard}} - 1 \equiv q - 1$, $r$ is the isotope ratio (unitless), z the depth (m), T the absolute temperature (K), $\Delta m$ the isotope mass difference (kg mol$^{-1}$), $g$ the gravitational acceleration (m s$^{-2}$), and R the fundamental gas constant (J mol$^{-1}$ K$^{-1}$).

Gradual accumulation of snow leads to a slow, downward advection of the enclosed air. The net air advection velocity is slower than the snow accumulation rate (yet still downward in the horizontal average in an Eulerian framework) because compression of the porous firn medium produces a return flow of air from the firn column to the atmosphere (Rommelaere et

al., 1997).

Buoyancy-driven convection and brief pressure anomalies associated with wind blowing over an irregular topography cause strong mixing between the near-surface firn and the atmosphere, smoothing out any concentration gradients (Colbeck, 1989; Kawamura et al., 2013; Severinghaus et al., 2010). Such convective mixing is usually observed only in the top few meters of the firn column. Convective mixing causes a deviation from the gravitational settling equilibrium (i.e., the solution

to Eq. 1) and leads to kinetic isotope fractionation because faster diffusing isotopes more easily overcome the same amount of mass-independent mixing (Buizert et al., 2012; Kawamura et al., 2013).

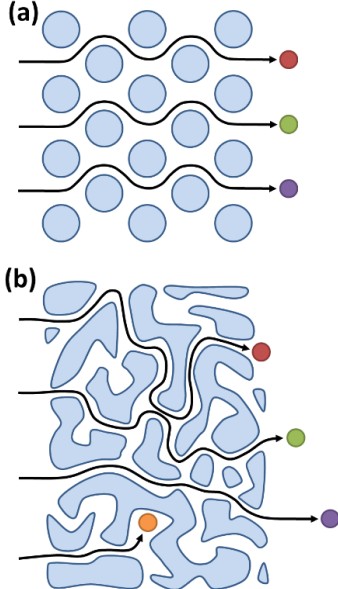

**Figure 2.** Schematic of (a) a non-dispersive medium and (b) a highly dispersive medium such as the deep firn. Figure reproduced from Buizert and Severinghaus, (2016).



Last, surface pressure variability on longer timescales (> 1 hour) drives air movement down to the firn-ice transition. Building on work by Schwander et al. (1988), Buizert and Severinghaus (2016) suggest that migrating storm centres may produce significant pressure gradients and induce fast airflows in firn. Porous firn has a high tortuosity, i.e., two points are typically connected by strongly curved paths, and the deep firn also contains many cul-de-sacs (Fig. 2). Airflow through such a medium produces mass-independent, dispersive mixing. Dispersion in this context is an emergent macroscopic phenomenon that describes microscopic deviations of the flow from Darcy's law of bulk fluid flow through porous media (Buizert and Severinghaus, 2016). This process may be accounted for by adding additional dispersion to the governing equation of trace gas transport.

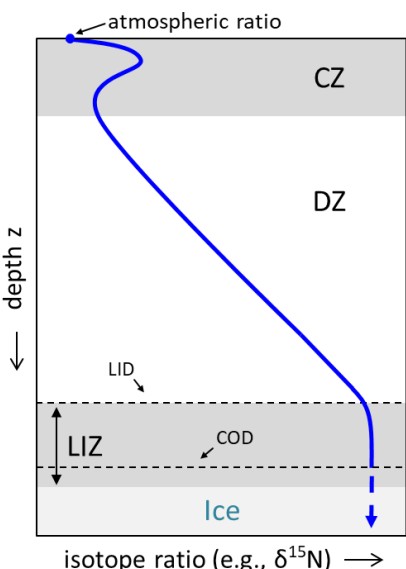

**Figure 3.** Schematic depiction of a typical isotope profile. The convective zone (CZ), the diffusive zone (DZ), the lock-in zone (LIZ) and the ice below are indicated by shading. The top of the lock-in zone is marked by the lock-in depth (LID). The close-off depth (COD) occurs where the air content becomes fixed and pressure in open porosity increases above hydrostatic pressure.

Together, these four processes yield a firn column that is typically split into a convective zone (CZ), a diffusive zone (DZ) and a lock-in zone (LIZ) (Fig. 3). The close-off depth (COD) occurs where the air content becomes fixed and pressure in open porosity increases above hydrostatic pressure. The CZ is rather well-mixed with a trace gas composition similar to the atmosphere. Nevertheless, large seasonal thermal gradients can lead to isotopic fractionation which is only partially attenuated by convective mixing (Kawamura et al., 2013; Severinghaus et al., 2001). Molecular diffusion dominates in the DZ but the effective firn diffusivity decreases with depth to represent the hindering effect of tortuosity on diffusion. Throughout the DZ, gravitational settling leads to an enrichment of all isotopes heavier than air in proportion to their mass difference. The top of the LIZ, the somewhat ill-defined lock-in depth (LID) horizon, is commonly deduced from a rather sudden change in the slope of the $\delta^{15}N$, $CO_2$ or $CH_4$ profiles. Gravitational enrichment of isotopes ceases in the LIZ and isotope ratios remain constant.




We term any such deviation from gravitational equilibrium "disequilibrium" (without implying that such a situation is not in steady-state).

The physical mechanism responsible for the cessation of gravitational enrichment in the deep firn is still not fully understood. Since CFCs and other anthropogenic tracers have been detected in firn air measurements well below the depth expected from pure advection, it is clear that vertical transport by molecular diffusion or dispersion continues in the LIZ (Buizert et al., 2012; Buizert and Severinghaus, 2016; Severinghaus et al., 2010). However, no further gravitational settling of isotopes occurs in the LIZ as indicated by constant $\delta^{15}N$ values. Furthermore, the vertical transport in the LIZ appears to be at least to some degree mass- and diffusivity-dependent because the faster diffusing $CH_4$ advances further in the LIZ than the slower diffusing gases CFC-113 or $CO_2$ (Buizert et al., 2012). Therefore, transport in the LIZ cannot be explained by either mass-indiscriminate dispersive mixing or molecular diffusion alone. Most current 1D firn air models greatly reduce molecular diffusivity in the lock-in zone and simultaneously introduce a mass-independent mixing term to match measured trace gas profiles (Buizert et al., 2012). A physical mechanism to justify these numerical methods remains elusive. Buizert and Severinghaus (2016) introduce barometric pumping in a 1D firn model. The authors observed a significant effect of barometric pumping on the $\delta^{15}N$ profile but needed to invoke a highly idealized parametrization of firn layering to match observations due to the intrinsic difficulties of representing layers in 1D.

Here we explore the possibility that non-fractionating trace gas mixing in deep firn may be explained by discontinuous layers of zero diffusivity and barometric pumping. High density layers are empirically linked to low porosity, diffusivity and permeability, increasing the firn's tortuosity and forcing extensive horizontal transport. The influence of layering and horizontal inhomogeneity on firn gas transport is mostly untested in numerical models so far since previous firn air models were generally limited to one dimension. In particular, we will test two mechanisms by which density layering could influence isotope ratios in firn air: Layering may (i) reduce gravitational settling of isotopes because the driving force for gravitational settling is effectively zero during horizontal transport and layering may (ii) modulate the mass-independent dispersive mixing effect of barometric pumping. Our analyses will focus on two Antarctic high-accumulation sites, WAIS Divide and Law Dome DSSW20K (Battle et al., 2011; Trudinger et al., 2002).

## 2 Methods

### 2.1 Governing equation and firn properties

2D trace gas transport in firn is simulated by numerically solving the advection-diffusion-dispersion equation, known from hydrology (Freeze, R.A., Cherry, 1979), adapted to firn,

$$\tilde{s}\frac{\partial q}{\partial t} = \vec{\nabla} \cdot \left[ \tilde{s}\boldsymbol{D}_m \left( \vec{\nabla}q - \frac{\Delta m\,\vec{g}}{R\,T}\,q + \Omega\frac{\partial T}{\partial z}q\,\hat{k} \right) + \tilde{s}\boldsymbol{D}_d\vec{\nabla}q \right] - (\tilde{s}\,\vec{u})\cdot\vec{\nabla}q \qquad (2)$$





with $q \equiv \delta + 1$ the ratio of any isotope to $^{28}N_2$ compared to a standard material, $\tilde{s} \equiv s_{op}\exp\left(\frac{\Delta m g z}{RT}\right)$ the pressure-corrected open porosity (m³ m⁻³), $T$ temperature (K), $\Omega$ thermal diffusion sensitivity (K⁻¹), and $\vec{u}$ the advection velocity (m s⁻¹). $\boldsymbol{D}_m$ and $\boldsymbol{D}_d$ are the 2D molecular diffusion and dispersion tensor (m² s⁻¹). $\vec{\nabla}q$ is the concentration gradient and $\vec{\nabla}\cdot$ denotes the 2D divergence operator. From left to right, the terms of Eq. (2) represent Fickian diffusion, gravitational fractionation, thermal

fractionation, dispersive mixing and advection. Since Eq. (2) is only valid for the binary diffusion of a trace gas into a major gas, ratios of any two isotopes of masses $x$ and $y$ are obtained by separately simulating the diffusion of each isotope into the major gas $^{28}N_2$ and using the relationship

$$q_{x/y} = \frac{q_{x/28}}{q_{y/28}} \tag{3}$$

to calculate the isotope ratios of interest (Severinghaus et al., 2010).

Isotope ratios are assumed to be constant at the surface (Dirichlet boundary) and reconstructions of atmospheric $CO_2$ and

$CH_4$ concentrations over the last 200 years are used to force runs of these anthropogenic tracers (see Supplementary Information (SI)). The bottom boundary is implemented by allowing only the advective flux to leave the domain (Neumann boundary). Diffusion and dispersive mixing cease below the COD. A periodic boundary condition is used in the horizontal direction. The horizontal extent of the model is varied between sites to maintain a constant ratio of annual layer thickness to the model's spatial extent. Firn density (Fig. 4a) is prescribed from a fit to the measured density profile at each site. Following

Severinghaus et al. (2010) and Kawamura et al. (2013), empirical relationships are used to derive open and closed porosities from the density profile (Fig. 4b). The pressure-corrected open porosity $\tilde{s}$ is assumed to be time independent.

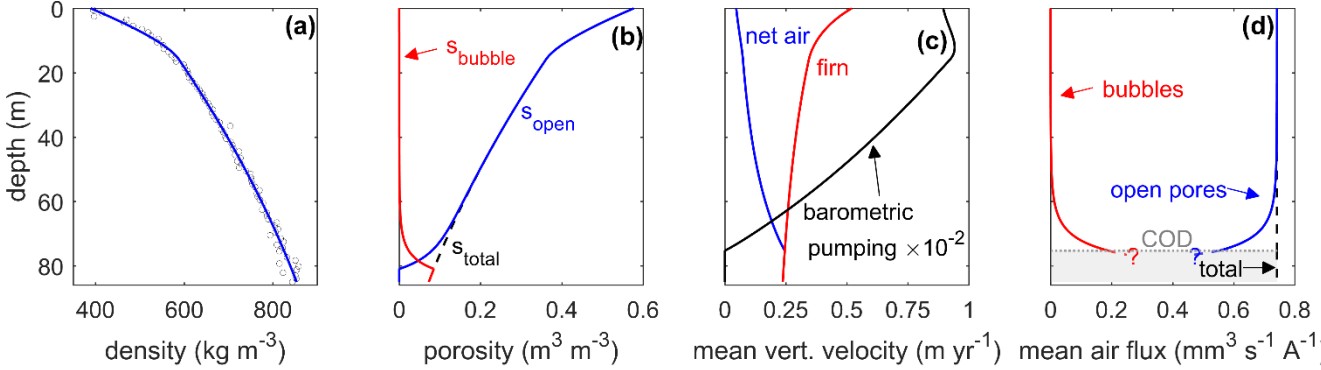

**Figure 4.** Firn conditions and modelled velocities profiles at WAIS Divide. (a) Density fit to observed data (data from Battle et al., 2011),
(b) open, bubble (i.e., closed) and total porosity, (c) horizontally averaged barometric pumping strength (i.e., time-mean horizontal average
of $|\vec{u}_b|$, black), and horizontally averaged net air ($\vec{w}_{firn} + \vec{u}_r$, blue) and firn velocity ($\vec{w}_{firn}$, red), and (d) mean air flux in open pores (blue)
and bubbles (red).




## 2.2 Advection velocity and barometric pumping

The 2D velocity field $\vec{u}$ is a result of a combination of (i) air migration with the firn $\vec{w}_{firn}$, (ii) return flow of air $\vec{u}_r$ from the firn to the atmosphere due to the gradual compression of pores, and (iii) airflow resulting from barometric pumping $\vec{u}_b$ (Figs. 4c & 5). Details on their derivation are provided in the SI. In short, $\vec{w}_{firn}$ is the vertical advection of snow and air in the firn

column and is constrained by assuming a constant snow and ice mass flux at all depths. The return flow $\vec{u}_r$ is calculated based on the effective export flux of air in open and closed pores at the close-off depth (COD), imposing a constant mean vertical air flux throughout the firn column (Fig. 4d) (Rommelaere et al., 1997; Severinghaus and Battle, 2006). Finally, the barometric pumping flow $\vec{u}_b$ is the airflow needed to re-establish hydrostatic balance in the firn in response to any surface pressure anomaly. Surface pressure variability is represented by pseudo red noise, mimicking observed pressure variability at both sites.

The near-coast location Law Dome is more strongly affected by storm activity than WAIS Divide with pressure variability $\sim$11.2 hPa day$^{-1}$ compared to $\sim$7 hP day$^{-1}$ at WAIS Divide. $\vec{u}_r$ and $\vec{u}_b$ follow Darcy's law of flow through porous media (Darcy, 1856):

$$\vec{u} = -\frac{\kappa}{\tilde{s}\,\mu}\nabla P, \tag{4}$$

with $\nabla P$ the pressure gradient, $\kappa$ the permeability of firn, and $\mu$ the viscosity of air (Fig. 5).

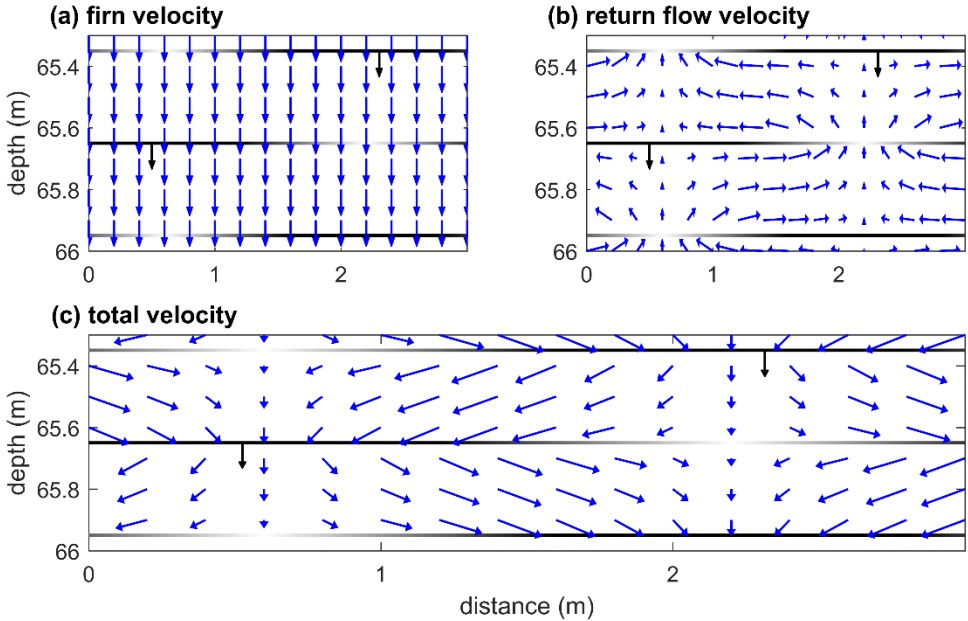

**Figure 5.** The different components of the velocity field. A linear combination of (a) the firn velocity and (b) the velocity of air return flow to the atmosphere due to pore compression yield (c) the net firn air advection velocity used in the model for a certain layer configuration. Because of its alternating direction, barometric pumping yields no net flow but instantaneous flow field patterns look similar to panel (b). Black arrows indicate the slow downward advection of layers at the firn velocity.



### 2.3 Firn layering

Idealized firn layering is implemented by forcing the vertical velocity components $u_r$ and $u_b$, as well as all vertical diffusive fluxes between the grid boxes on either side of a layer to be zero. Only the advection of air with the firn ($\vec{w}_{firn}$) remains active for these grid box boundaries. Layering limits vertical gas transport and yields almost exclusively horizontal transport between layers. Here, we represent layers to have infinitesimal thickness, because of the computationally limited spatial resolution of the model. Layers are repeatedly introduced at a specific depth and migrate down with the velocity of the firn. The vertical distance between layers is set to correspond to the snow accumulation of one year and the horizontal extent of layers increases linearly with depth until they cover the entire domain at the close-off depth (COD). The mean layer opening size is held proportional to the annual layer thickness at all sites to make the vertical advection velocities independent of the arbitrary horizontal extent of the model. To obtain more realistic flow fields, the permeability of layers tapers off at both ends of a layer.

### 2.4 Dispersive mixing

The dispersion tensor $\boldsymbol{D}_d$ is made up of two components, (i) convective mixing of air in a shallow region near the surface and (ii) dispersive mixing caused by barometric pumping in the more tortuous, deep firn. First, the well-mixed convective zone is commonly represented by mass-independent diffusion acting in the vertical and horizontal directions. The diffusivity profile of convective mixing $D_c = D_c(z)$ is described as an exponential decay away from the snow-atmosphere interface (Kawamura et al., 2013). Its maximum surface value and the decay constant are chosen to match observed $\delta^{15}$N values in the deep firn. After reaching a specified maximum depth of 8 m or 14 m at WAIS Divide and Law Dome DSSW20K, respectively, the diffusivity tapers linearly to zero over 2 m. Second, airflow through any dispersive medium leads to longitudinal-to-flow (i.e., along the flowline) and transverse-to-flow (i.e., across the flowline) mixing. Because barometric pumping velocities are orders of magnitude faster than the return flow, dispersive mixing is dominated by barometric pumping. The 2D dispersion tensor becomes:

$$\boldsymbol{D}_d = \begin{bmatrix} D_L \frac{u_n^2}{v^2} + D_T \frac{w_n^2}{v^2} + D_c & \frac{u_n w_n}{v^2}(D_L - D_T) \\ \frac{u_n w_n}{v^2}(D_L - D_T) & D_T \frac{u_n^2}{v^2} + D_L \frac{w_n^2}{v^2} + D_c \end{bmatrix}, \tag{5}$$

where $u_n \equiv u_r + u_b$ and $w_n \equiv w_r + w_b$ are the sum of the return and barometric pumping velocity components in the x- and the y-directions with $v \equiv (u_n^2 + w_n^2)^{0.5}$. $D_L$ and $D_T$ are the longitudinal and transverse dispersion coefficient (m$^2$ s$^{-1}$), respectively. $D_L$ and $D_T$ are commonly approximated as linear functions of velocity (Freeze, R.A., Cherry, 1979)

$$D_L = \alpha_L v \tag{6}$$

$$D_T = \alpha_T v \tag{7}$$





where the proportionality factors $\alpha_L$ and $\alpha_T$ are the longitudinal- and transverse-to-flow dispersivity (m).

The degree of dispersive mixing in firn presumably depends on the direction of flow and probably also differs between the longitudinal-to-flow and transverse-to-flow direction. However, the treatment of anisotropic media is complex and only one parametrization for vertical, longitudinal-to-flow dispersion in firn is currently available (Buizert and Severinghaus, 2016).

Therefore, we assume that the dispersivity $\alpha$ of firn is isotropic (i.e., $\alpha_L = \alpha_T \equiv \alpha$). This simplifies $\boldsymbol{D}_d$ to

$$\boldsymbol{D}_d = (\alpha \, v + D_c) \, \boldsymbol{I}, \tag{8}$$

with $\boldsymbol{I}$ the second order identity matrix. The dispersion flux term in Eq. (2) becomes

$$\tilde{s} \boldsymbol{D}_d \vec{\nabla} q = \tilde{s} \, [\alpha \, v + D_c] \, \vec{\nabla} q. \tag{9}$$

The dispersivity parametrization of Buizert and Severinghaus (2016) is based on direct measurements of cylindrical firn samples from Siple Station, Antarctica, performed by Schwander et al. (1988). The parameterization relates dispersivity to open porosity $s_{op}$ as

$$\alpha(s_{op}) = \tilde{s} \left[ 1.26 \cdot \exp(-25.7 \, s_{op}) \right]. \tag{10}$$

Here, the factor of $\tilde{s}$ was added to the original parameterization by Buizert and Severinghaus (2016) because $\alpha$ relates dispersive mixing to the velocity components $u_n$ and $w_n$, that denote flow velocities per unit pore cross-section ($w_{pores}$). Schwander et al. (1988), however, originally measured the considerably slower bulk airflow per unit firn cross-section (i.e., $w_{bulk} = \frac{w_{pores}}{\tilde{s}}$). Since dispersivity is a scale dependent property, it is important to use parametrizations that are compatible with the resolution of the numerical model. The sample size of Schwander et al. (1988) (i.e., 30 mm diameter & 50 mm length)

approximately matches the resolution of our numerical model (i.e., 30 x 40 mm) and thus should adequately approximate subgrid-scale (i.e., pore-scale) mixing processes that currently cannot be resolved. Spatial inhomogeneity of subgrid-scale firn dispersivity that was not captured by the sampling of Schwander et al. (1988) cannot be accounted for in the model. Dispersion on larger scales such as the interaction of flow and layers is explicitly represented in the model by the interplay of advection and diffusion. Thus, dispersive mixing is completely constrained in the model and based on empirical parameterizations that

are not subject to any tuning.

## 2.5  Molecular diffusion

The (effective) molecular diffusivity profile is established by simultaneously fitting the simulated $CO_2$ and $CH_4$ profile to real firn measurements at both sites. Effective vertical diffusivity decreases with depth to represent the subgrid-scale effect of decreasing pore connectivity and increasing firn tortuosity. A spline function defines the effective vertical diffusivity profile

which decreases monotonically from the surface to zero at the COD (Fig. 6). Diffusivities for other trace gases are calculated



by scaling the tuned $CO_2$ diffusivity by the free air diffusivity of each gas relative to $CO_2$ (Trudinger et al., 1997). The diffusivity tuning presents an underconstrained problem because horizontal and vertical molecular diffusivities are essentially free parameters. It is qualitatively evident from firn air sampling that horizontal connectivity/diffusivity is much higher than vertical diffusivity in the deep firn but this observation is incompletely quantified. Here, horizontal molecular diffusivities are

fixed to 10x the vertical diffusivity at the same depth and annual layers represented some fraction of the total tortuosity explicitly in the model. There are many degrees of freedom in tuning molecular diffusivities and the diffusivity parameterization is therefore not unique.

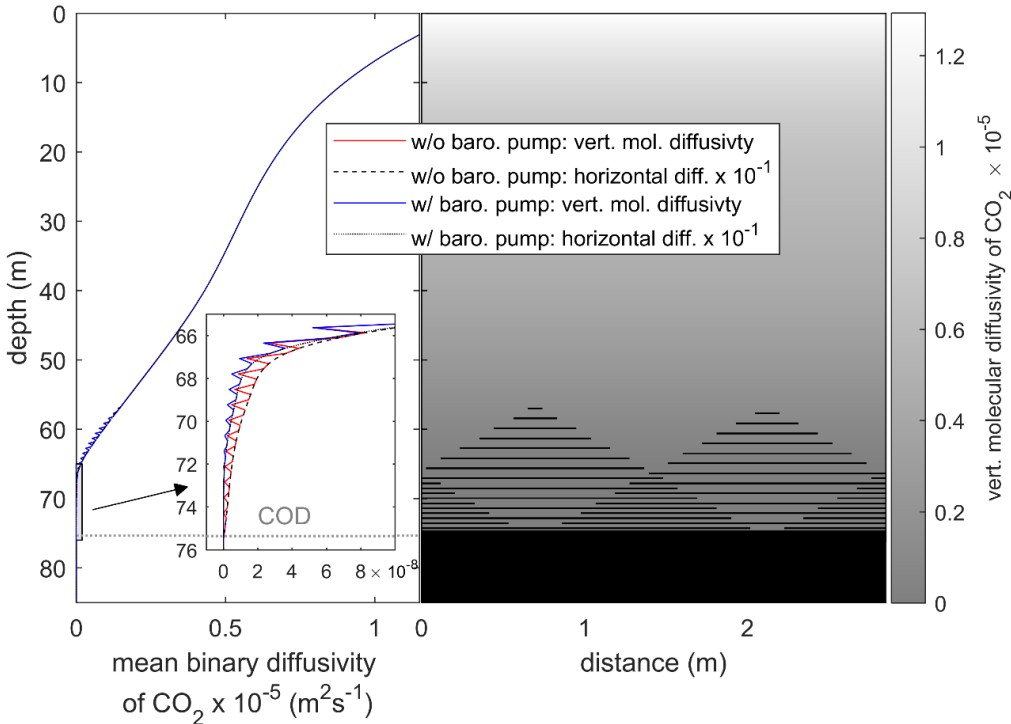

**Figure 6.** The $CO_2$ diffusivity profile at WAIS Divide. Left panel: Horizontally averaged, vertical and horizontal diffusivity in the model
with and without barometric pumping. Right panel: map of diffusivity in the 2D model without barometric pumping. Only every third layer present in the model is shown here for clarity.

## 2.6    Thermal fractionation and the temperature model

Finally, a 1D thermal model by Alley and Koci (1990) is run separately to simulate the temperature evolution of the firn. The model is forced by a long-term surface temperature trend based on published records by Van Ommen et al. (1999), Dahl-
Jensen et al. (1999) and Orsi et al. (2012). A generic Antarctic seasonal cycle derived from a ~8-10-year climatology of automatic weather station observations at WAIS Divide and Law Dome (Lazzara et al., 2012) is superimposed on this trend. Alley and Koci's temperature model is based on the heat transport equation for firn (Johnsen, 1977) with parametrizations for firn thermal properties from Weller and Schwerdtfeger (1977). Horizontal temperature gradients in firn are small at both sites and neglected in this study.





Considerable temperature gradients can exist in present-day firn because of recent global atmospheric warming and these gradients can lead to increased isotope thermal fractionation, in particular of $\delta^{15}N$. The sensitivity of isotopes to diffuse in response to temperature gradients is captured by the thermal diffusion sensitivity $\Omega$. Values of $\Omega$ are approximated as

$$\Omega = \frac{a}{T} - \frac{b}{T^2} \tag{11}$$

or assumed to be temperature independent if the temperature sensitivity is unknown (Severinghaus et al., 2001). Coefficients

$a$ and b were determined experimentally for different isotope ratios by Grachev and Severinghaus (2003a, 2003b), Kawamura et al. (2013) and Kawamura (unpublished).

A detailed overview of further model parameters and the numerical realization of 2D gas transport may be found in the SI (SI Sect. 1-4).

## 3    Results

### 3.1    WAIS Divide

#### 3.1.1    CO₂ and CH₄

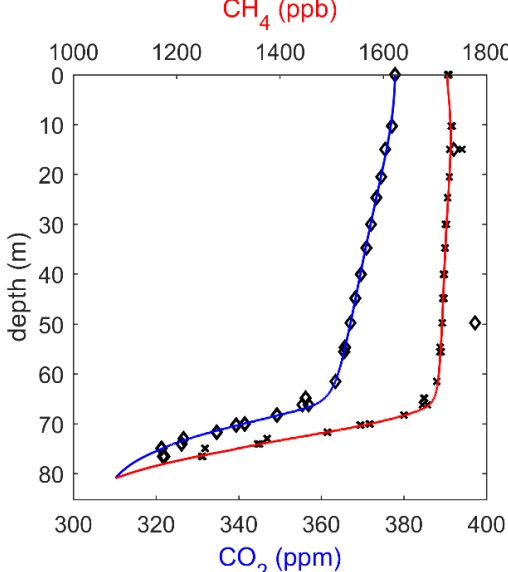

**Figure 7.** Simulated and observed $CO_2$ and $CH_4$ concentrations in the firn at WAIS Divide. The model is initialized with the recorded atmospheric trace gas concentrations in 1800 CE at all depths and is forced at the surface with histories of atmospheric $CO_2$ and $CH_4$

concentrations (Buizert et al., 2012; Dlugokencky et al., 2016a, 2016b, Etheridge et al., 1996, 1998; Keeling et al., 2001). Markers indicate observed $CO_2$ (diamonds) and $CH_4$ (crosses) concentrations (Battle et al., 2011). Differences in the $CO_2$ and $CH_4$ profiles between the 1D model and the 2D model with or without barometric pumping are not visible at the resolution of this figure but are illustrated in the SI (Fig. S9).



A comparison of simulated and observed $CO_2$ and $CH_4$ profiles shows good agreement at WAIS Divide, supporting the plausibility of our layered diffusivity parameterization (Fig. 7). In line with observations, both $CO_2$ and $CH_4$ concentrations decrease slowly with depth until ~68 m below the surface where they begin to decrease more rapidly.

In the following we will examine and compare results from four different permutations of the 2D model with or without
impermeable layers and with activated or deactivated barometric pumping. In versions without layering, our 2D model loses all horizontal heterogeneity and will thus be referred to as a '1D model' throughout the text. Since the explicitly implemented tortuosity from layering in the 2D model affects molecular diffusion and dispersion equally it is represented by equally lowering the effective molecular diffusivity and dispersivity in the layered region of the 1D version instead. Diffusivities are tuned such that the $CO_2$ profiles are (nearly) identical. The small remaining deviations in $CO_2$ and $CH_4$ concentrations between
model permutations (< ±0.8 ppm and < ±5 ppb, respectively) are illustrated in the SI (Fig. S9).

### 3.1.2    δ15N and thermal fractionation

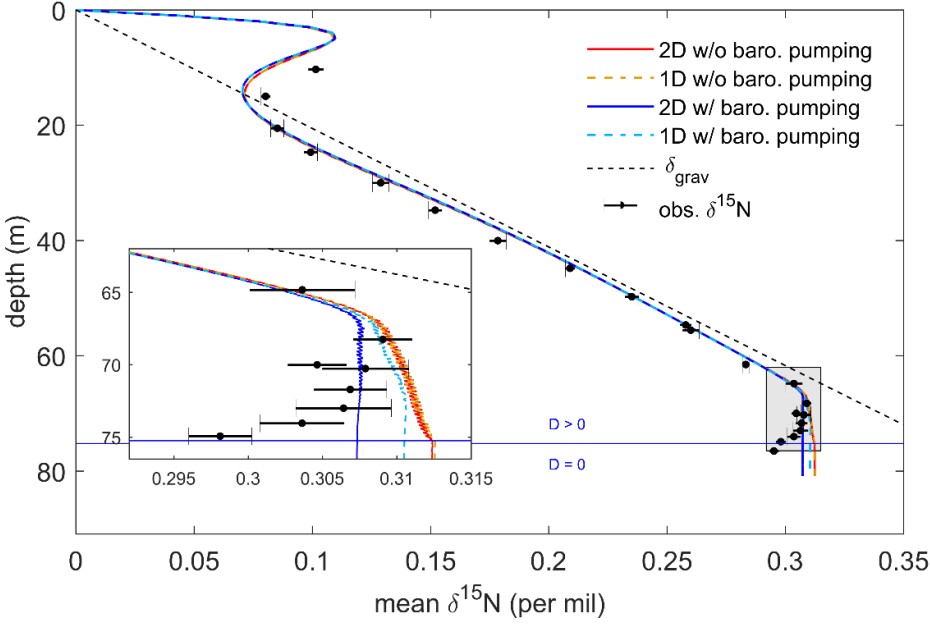

**Figure 8.** Horizontally averaged $\delta^{15}N$ at WAIS Divide. Model output is shown from four different versions of the 2D model (see text). Black circles with error bars indicate the observed firn $\delta^{15}N$ (Battle et al., 2011). The dashed black line represents the equilibrium solution for pure
gravitational settling ($\delta_{grav}$). The horizontal blue line marks the depth where vertical diffusivity reaches zero. The inlay shows a magnification of the lock-in zone.

In all four model setups, the seasonal cycle of temperature dominates the shape of the $\delta^{15}N$ profiles in the top ~30 m (Fig. 8). Warm summer temperatures drive the migration of heavy isotopes into the colder firn below and produce a $\delta^{15}N$ summer peak just below the surface (Severinghaus et al., 2001). In contrast, a minimum of $\delta^{15}N$ occurred in this region during the previous
winter season when the thermal gradients were reversed. The remnants of this winter minimum are still visible in the gas record as anomalously low $\delta^{15}N$ values below the summer peak. The differences between observations and simulated $\delta^{15}N$ values in



the top of the firn column are likely caused by the idealized representation of the seasonal cycle in the model. Anomalous temperature gradients associated with extraordinary weather events just before firn air sampling will modify the observed $\delta^{15}N$ at the site but are unaccounted for in the model.

### 3.1.3 Impact of (near-) impermeable layers

In the layered 2D model without barometric pumping, the simulated $\delta^{15}N$ values are close to observations at the top of the LIZ but continue to increase with depth (Fig. 8). This is contradicted by observational data even when the unusually low $\delta^{15}N$ values right on the COD and below are not taken into consideration (near the COD, firn air pumping becomes more difficult in the field and the potential for fractionation during sampling is increased). A closer inspection of the lock-in zone in Figs. 8 and 9 reveals a zigzag pattern in the $\delta^{15}N$ profile where impermeable layers are present. Isotope ratios are higher just above

horizontal layers where heavy isotopes can accumulate and are anomalously low below layers where they are more readily removed than supplied by gravitational settling. Gravitational settling through gaps in the layers sets up small horizontal concentration gradients that drive horizontal Fickian diffusion. Layering increases the effective travel path length for molecules and reduces the effective vertical diffusivity by increasing the tortuosity. However, layering alone appears to be insufficient to prevent gravitational settling completely because continued gravitational enrichment is observed in the LIZ in this model

version.



**Figure 9.** Simulated $\delta^{15}N$ in a part of the lock-in zone at WAIS Divide from the 2D model not including barometric pumping. Impermeable horizontal layers are shown in red. The openings sizes in the layers shrink with increasing depth.

Is the impact of layers on the firn trace gas profile larger for isotopes such as $\delta^{15}N$ than for anthropogenic tracers such as

$CO_2$, CFCs, or $CH_4$? All three gases have experienced large atmospheric variability in the industrial era. Therefore, the migration of these gases into the firn is dominated by vertical and horizontal Fickian diffusion in contrast to $\delta^{15}N$ where the



effect of gravity is critical for transport. To answer the above question, we compare output from the layered 2D model to the 1D model without layers. We find that $\delta^{15}N$ values in the 1D and 2D model without barometric pumping are almost identical (Fig. 8). Layering in the 2D model increases the effective transport distance of $CO_2$ just as much as for $\delta^{15}N$ and there is no disproportional impact of layering on gravitationally fractionated isotope ratios. Differences in explicitly represented tortuosity

are automatically compensated in the 1D model during tuning to the same $CO_2$ profile by reducing molecular diffusivities. Therefore, we conclude that layering alone cannot simultaneously explain the observed $CO_2$ and $\delta^{15}N$ patterns.

### 3.1.4   Barometric pumping and the emergence of the LIZ

$\delta^{15}N$ values simulated by the 1D and 2D model with barometric pumping are lower in the LIZ than in both model versions without barometric pumping (Fig. 8). Accounting for barometric pumping improves the agreement with observations

throughout the lock-in zone. However, the reduction of gravitational fractionation is substantially stronger when layers are present. Only when both layering and barometric pumping are accounted for in the model simultaneously, does the $\delta^{15}N$ profile correctly indicate no further gravitational enrichment in the LIZ and closely matches observations. Dispersive mixing is independent of molecular mass and does not lead to gravitational fractionation, but rather acts to eliminate the concentration gradients associated with gravitational settling. Although barometric pumping velocities are largest near the surface (Fig. 4c),

significant dispersive mixing is generally limited to the LIZ because the dispersivity of firn is inversely related to the open porosity in the parameterization of Buizert and Severinghaus (2016) and dispersion is overwhelmed by molecular diffusion in the DZ. Furthermore, molecular diffusivities drop rapidly in the LIZ when barometric pumping is active in the model (Fig. 6). Because dispersion provides an additional transport mechanism for trace species, less molecular diffusion is needed to match observed $CO_2$ and $CH_4$ concentrations. Layering amplifies the importance of barometric pumping because gravitational

fractionation between annual layers is restricted into the small gaps in the LIZ (Fig. 9). A lack of alternative pathways amplifies barometric pumping flows and thus dispersive mixing in these regions (Fig. 5), overwhelming the influence of gravitational fractionation more readily than in the 1D model. This effect is responsible for the large differences between the $\delta^{15}N$ profiles obtained from the two models with barometric pumping in Fig. 8. The strength of dispersive mixing in our layered 2D model is physically motivated; thus, barometric pumping and layering together lead to a more natural emergence of the lock-in zone

in the 2D model.

### 3.1.5   The convective zone height

We estimate the depth of the convective zone at WAIS Divide to be ~2.8 m. Multiple different procedures have been used to estimate convective zone heights in the past, many of which rely on $\delta^{15}N$ data in the deep firn near the LIZ (Battle et al., 2011). However, if the deep firn is affected by dispersive mixing due to barometric pumping, these estimates may be falsely attributing

some fraction of the dispersive mixing in the deep firn to the convective zone. To address this problem, we follow the method of Severinghaus et al. (2010) in calculating convective zone thickness. This approach compares the depth where $\delta^{15}N$ reaches a certain value in two different model configurations with and without convection. Thermal effects are neglected. The first



setup is the 2D model with barometric pumping as presented above but the dispersivity is set to zero everywhere without retuning the model. The convective zone height is calculated from the depth difference between this model run and a second model run where barometric pumping and the convective zone are deactivated and only advection and gravitational fractionation shape the profile of $\delta^{15}N$. Our height estimate is within the range of values from 1.4 to 5.2 m published previously

(Battle et al., 2011).

### 3.2 Law Dome DSSW20K

At Law Dome DSSW20K, the firn thickness is ~20 m less than at WAIS Divide. Accumulation rates are comparable, but temperatures are ~10 K warmer. The convective zone is slightly deeper and barometric pumping is stronger at Law Dome, yielding more convective and dispersive mixing. Constraining the convective zone height at DSSW20K is complicated because

less $\delta^{15}N$ measurements are available for this site, and their associated uncertainty is, at $\pm 15$ per meg, much larger than at the more recently sampled WAIS Divide site. Molecular diffusion generally takes a less important role at DSSW20K and molecular diffusivities obtained by tuning are about half or less than those at WAIS Divide for most of the firn column. Thermal fractionation has a weaker impact on the isotope record near the surface at Law Dome due to the smaller amplitude of the seasonal cycle and stronger convective mixing compared to WAIS Divide. Figures of molecular diffusivity, advection

velocities and other firn properties at site DSSW20K are available in the SI Sect. 5.

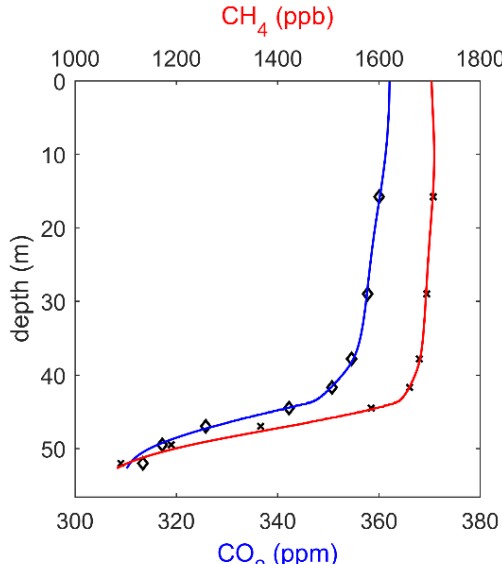

**Figure 10.** Simulated CO2 and CH4 concentrations in the firn at Law Dome DSSW20K. The model is forced with histories of atmospheric $CO_2$ and $CH_4$ concentrations from 1800 to 1998 CE (the date of sampling). Markers indicate observed $CO_2$ (diamonds) and $CH_4$ (crosses) concentrations (Trudinger et al., 2002).




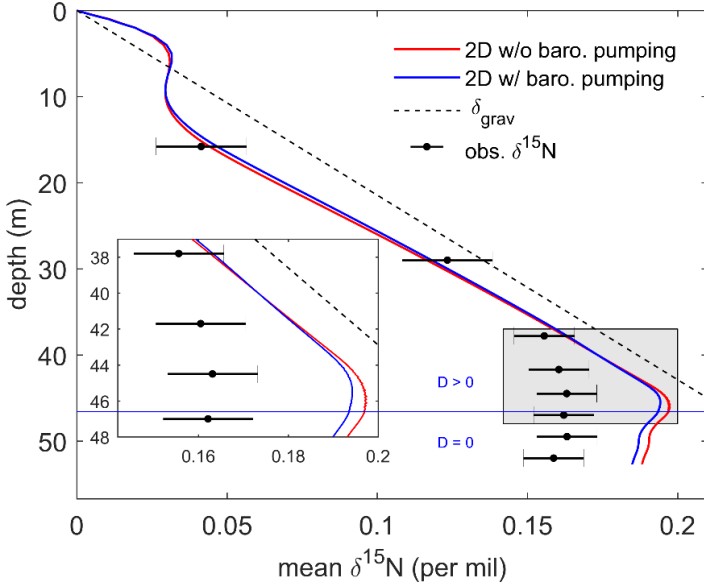

**Figure 11.** Horizontally averaged $\delta^{15}N$ at Law Dome DSSW20K. The solid blue and red line show the results of the 2D model including and excluding barometric pumping, respectively. Black circles with error bars indicate the observed firn $\delta^{15}N$ (Trudinger et al., 2002, 2013). The dashed black line represents the equilibrium solution for pure gravitational settling ($\delta_{grav}$). The horizontal blue line marks the depth where vertical diffusivity reaches zero. The inlay shows a magnification of the lock-in zone.

Simultaneously matching the $\delta^{15}N$, $CO_2$ and $CH_4$ profile at Law Dome DSSW20K has proven difficult in the past (Buizert and Severinghaus, 2016; Trudinger et al., 2002). Simulated $\delta^{15}N$ in the LIZ is typically significantly higher than in observations. Buizert and Severinghaus (2016) suggested that barometric pumping in the deep firn may be able to reconcile this contradiction. However, the mixing obtained from theoretical predictions was insufficient to obtain the anticipated results. Buizert and Severinghaus (2016) hypothesized that firn layering may play a critical role in amplifying the impact of barometric pumping. They used an idealized eddy and molecular diffusivity profile in the deep firn to simulate the effect of layers on firn air transport. Using these diffusivity profiles, they were able to obtain good agreement with observed $\delta^{15}N$, $CH_4$ and $^{14}CO_2$. Our 2D model includes an explicit representation of layering and places similar physical constrains on barometric pumping as the 1D model of Buizert and Severinghaus (2016). The model is tuned to optimize agreement with $CO_2$ and $CH_4$ and the patterns of both profiles are reproduced correctly (Fig. 10). But the disagreement between modelled and observed $\delta^{15}N$ in the deep firn remains despite barometric pumping producing significant non-fractionating dispersive mixing in the region (Fig. 11). Simulated $\delta^{15}N$ values diverge from observations at ~38 m, where gravitational enrichment seems to stop in observations but continues in the model. The LIZ, as indicated by $CO_2$ and $CH_4$, in contrast only starts at roughly 43 m depth. Such an early onset of dispersive mixing is not supported by the dispersivity parameterization. However, only the longitudinal-to-flow mixing in the vertical direction at Siple Station was used to develop the firn dispersivity parameterization, and the use of this parameterization may be inappropriate at Law Dome DSSW20K (Buizert and Severinghaus, 2016). Moreover, dispersivity typically differs in the horizontal and vertical as well as the longitudinal- and the traverse-to-flow directions, an effect that is not accounted for in this study because of a lack of observational evidence to constrain anisotropic dispersivity.



## 4    Discussion

### 4.1    Differential kinetic isotope fractionation

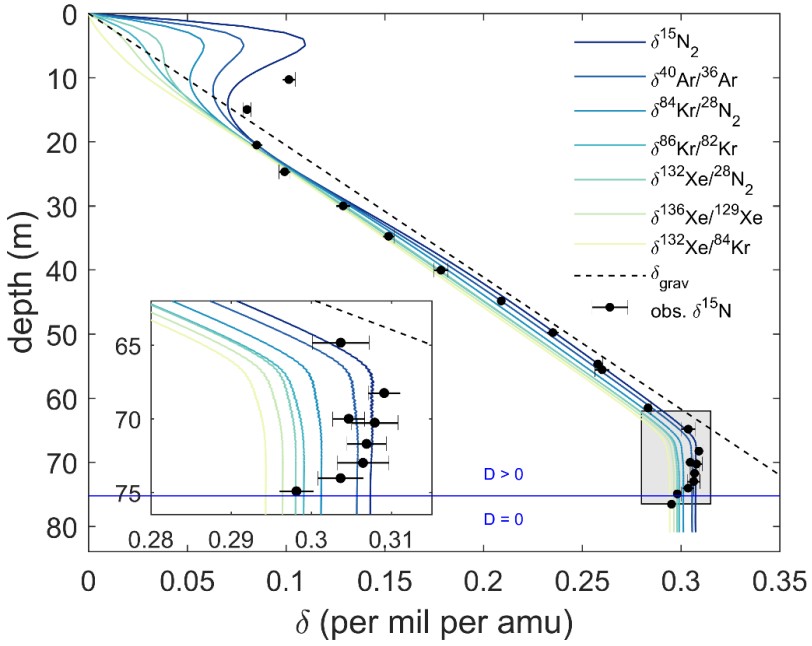

**Figure 12.** Horizontally averaged isotope ratios at WAIS Divide in the 2D model including barometric pumping. Isotope ratios are normalized to one atomic mass unit (amu, see SI) mass difference (SI Sect. 6). The dashed black line represents the equilibrium solution for pure gravitational settling ($\delta_{grav}$). The horizontal blue line marks the depth where vertical diffusivity reaches zero. Observed $\delta^{15}N$ are shown as circles with horizontal error bars (Battle et al., 2011). The inlay shows a magnification of the lock-in zone (grey patch).

Isotope ratios in firn typically do not reach values as high as predicted from gravitational equilibrium due to the influence of advection and non-fractionating dispersive mixing. Advection and mass-independent mixing transport less fractionated air down in the firn column and act to counterbalance the enrichment of heavy isotopes by gravitational fractionation. All isotope ratios fall below the gravitational settling line $\delta_{grav}$ in Fig. 12 but the magnitude of the deviation depends on the specific isotope pair. This difference between the mass-normalized Kr and Ar isotope ratios has been termed $^{86}Kr$ excess (Buizert and Severinghaus, 2016). At the COD of WAIS Divide, simulated $^{86}Kr$ excess is $\sim 5.6$ per meg per amu in the 2D model with barometric pumping. This is significantly lower than the $7 - 22$ per meg per amu values observed in the WAIS Divide ice core (Orsi A., personal communication).

The magnitude of disequilibrium of different isotope and elemental ratios is quantified by defining the (mass-normalized) kinetic fractionation relative to $\delta^{15}N$ ($\epsilon'$) (Kawamura et al., 2013) as

$$\epsilon'_{x/y} \equiv \frac{1}{1000 \times \Delta m_{x/y}} \ln\left(\frac{q_{x/28}}{q_{y/28}}\right) - \ln\left(q_{^{15}N}\right), \tag{12}$$




where $\Delta m_{x/y}$ is the mass difference of isotopes $x$ and $y$. This definition is similar to the $^{86}$Kr excess terminology introduced by Buizert and Severinghaus (2016) but $\epsilon'$ is given in the more precise ln(q)-notation and uses $\delta^{15}$N as the reference instead of $\delta^{40}$Ar/ $\delta^{36}$Ar. Here, isotope ratios are assumed to be corrected for the influence of thermal fractionation either through combined Ar and N$_2$ measurements on firn air (Grachev and Severinghaus, 2003b, 2003a) or, as done here, by removing temperature

effects with a suitable firn air transport model (Fig. 13).

$\epsilon'$ is caused by differential kinetic isotope fractionation. Heavy, slow-diffusing isotopes approach gravitational equilibrium more slowly than lighter, faster-diffusing isotopes. Therefore, ratios of heavier elements are more susceptible to kinetic fractionation. Consequently, $\epsilon'$ is more negative for heavier, slower diffusing isotopes. On its own, this simple rule of thumb cannot explain the pattern of ratios containing two different elements, such as $^{132}$Xe/$^{28}$N$_2$. The magnitude of disequilibrium in

such mixed-element ratios is further discussed in Appendix B.

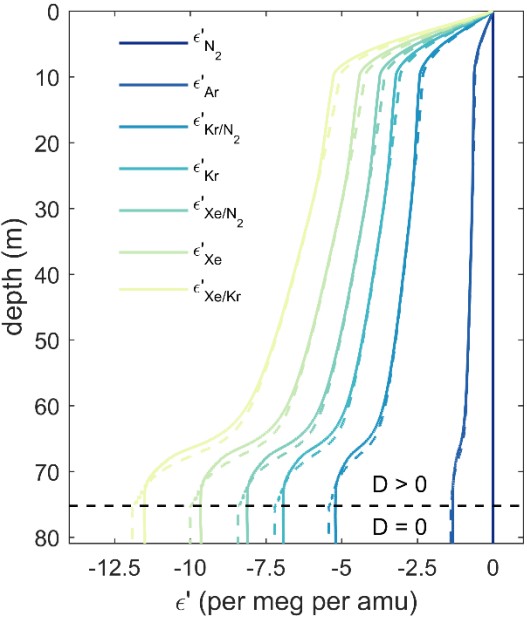

**Figure 13.** Differential kinetic isotope fractionation ($\epsilon'$) profiles for different isotope pairs at WAIS Divide. Coloured solid and dashed lines show results from the 2D model with and without barometric pumping, respectively. $\epsilon'$ is defined as the (typically negative) difference between any mass-normalized isotope ratio and $\delta^{15}$N as detailed in the text. Subscripts of one or two element names identify ratios as isotope 15 or elemental ratios, respectively. The dashed black line highlights where molecular diffusivity in the model reaches zero.

In the DZ $\epsilon'$ decreases almost linearly with depth, while within the convective zone and in the LIZ $\epsilon'$ changes much more rapidly. Where molecular diffusivity is zero, $\epsilon'$ remains constant. This pattern is explained by the relative importance of advection and dispersive mixing compared to molecular diffusion in the firn column. The vertical Péclet number (Pe) traditionally quantifies the ratio of the advective to the diffusive transport and is defined as the ratio of the diffusive to the

20 advective time scale. Here we add the time scale of dispersive mixing to the numerator because the effect of advection and dispersive mixing on the isotope profiles is very similar although the physics differ (Kawamura et al., 2013).




$$\text{Pe} \equiv \frac{\tau_{adv} + \tau_{D_e}}{\tau_{D_m}} = \frac{\frac{W}{L} + \frac{D_e}{L^2}}{\frac{D_m}{L^2}} = \frac{WL + D_e}{D_m},$$

(13)

where $L = 1$ m is the characteristic length scale of firn air transport, and $D_m, W$ and $D_e$ are characteristic values of the molecular diffusivity, the time mean vertical advection velocity, and the vertical dispersive or convective mixing at that depth, respectively.

The Péclet number calculated in the model for the WAIS Divide varies by many orders of magnitude through the firn
column with peak values in the convective zone and the deep firn (Fig. 14). High Péclet numbers in the convective zone are caused primarily by large $D_e$ values, and high Péclet numbers in the LIZ are mostly the result of low molecular diffusivities. Kawamura et al. (2013) showed analytically that kinetic isotope fractionation depends on the ratio of eddy diffusivity to molecular diffusivity, but the role of advection was neglected due to near-zero accumulation rates at the Megadunes site. The absolute difference in kinetic isotope fractionation (i.e. $\epsilon'$) should be greatest when the product of the Péclet numbers of both
isotopes is near one. In line with these theoretical predictions, we observe almost no further isotopic enrichment of $\delta^{15}$N in the lock-in zone when barometric pumping is included in the model and Pe $\gg$ 1 (Figs. 8 & 14). The largest changes of $\epsilon'$ occur in the 2D model when the Péclet number is within approximately 1-2 orders of magnitude of unity. The grey bar in Fig. 14 contains the convective zone as well as the region just above the LIZ where non-fractionating mixing is of similar magnitude as molecular diffusion.

With active barometric pumping and centimetre-scale layering, the product of the Péclet numbers at the bottom of the LIZ becomes so large that $\epsilon'$ stops to decrease entirely in our model. If barometric pumping is neglected instead, the Péclet numbers in the layered 2D model are considerably lower in the LIZ and some gravitational and kinetic fractionation persists (i.e., $\delta^{15}$N and $\epsilon'$ continue to change gradually). Therefore, barometric pumping leads to slightly weaker rather than stronger differential kinetic fractionation at the COD of WAIS Divide. Layering and barometric pumping in the model seem to be insufficient to
obtain the extreme values of $\epsilon'_{Kr}$ (or $^{86}$Kr excess), observed in the WAIS Divide ice core record. Instead, other, unresolved (i.e., subgrid-scale) process may be the reason for the large observed $^{86}$Kr excess. For example, on the pore level, advective flows may be channelled into wider pores because the hydraulic conductance scales with the fourth power of the pore radius, whereas the diffusive conductance only depends on the square of the pore radius as indicated by the Hagen–Poiseuille equation (Buizert and Severinghaus, 2016). On small scales, Fick's law may also not be the correct relation to represent the physical
process of diffusion. The mean square displacement in the disordered firn medium is not necessarily linearly depend on the product of diffusivity and time and we suggest other models of diffusion should be explored. Buizert and Severinghaus (2016) hypothesize that $^{86}$Kr excess may be primarily produced by barometric pumping in the deep firn and could thus be used as a measure of paleo storminess in ice core records. Our findings suggest that establishing a straightforward relationship between $^{86}$Kr excess and surface pressure variability using firn air models alone may not be possible without more observational data.





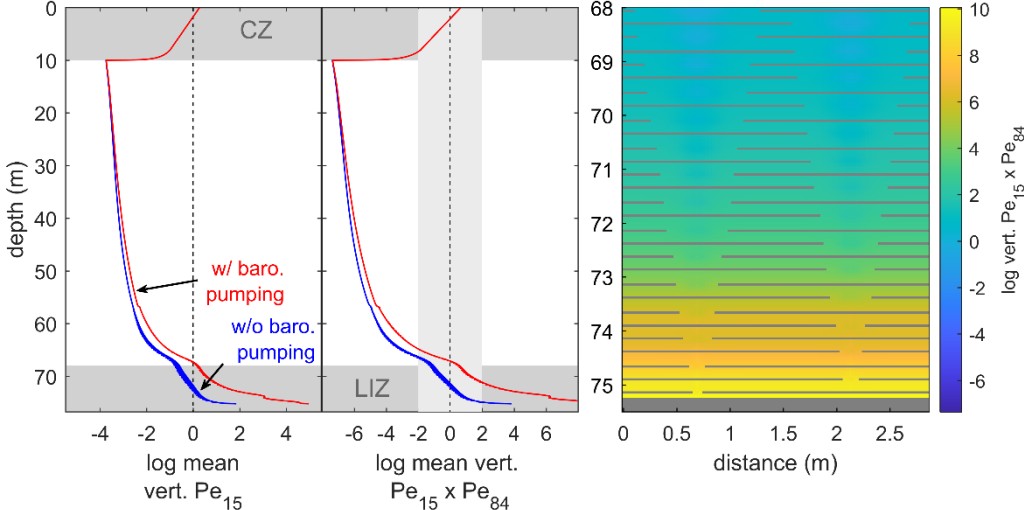

**Figure 14.** The balance of fractionating and non-fractionating mixing at WAIS Divide. The left panel illustrates the horizontally averaged Péclet number of $\delta^{15}N$ (Pe$_{15}$, see text). Blue and red lines show results from the 2D models with and without barometric pumping. The strength of dispersive mixing in the calculations is given by the mean barometric pumping flow velocities at the site. The middle panel displays the product of the Péclet numbers for $\delta^{15}N$ and $^{84}Kr/^{28}N_2$ (Pe$_{84}$). The region where $\epsilon'$ changes with depth should be greatest (i.e., where the product of the Péclet numbers is near one) is highlighted by a grey bar. The right panel provides a magnified 2D map of this Péclet number product in the LIZ. Note that the Péclet number becomes infinite in the region below the COD and at layers where molecular diffusivities are zero. These regions are not considered when taking horizontal averages and are coloured grey in the right panel for illustratory purposes.

## 4.2 Diffusive fractionation

Strong kinetic isotope fractionation was also observed for trace gases that experienced large changes in the atmospheric mixing ratio while the atmospheric isotope ratios remained constant (Buizert et al., 2013; Trudinger et al., 1997). As the concentration of a trace gas increases, the isotopologues of the gas migrate into the firn column at different speeds because of small differences in their masses and diffusivities. This results in a relative depletion of the slower diffusing isotopologue with depth called diffusive fractionation (Trudinger et al., 1997). During periods of abrupt CH$_4$ release or sequestration diffusive fractionation commonly amounts to a relevant correction in ice core studies (Buizert et al., 2013; Trudinger et al., 1997). Diffusive fractionation of $\delta^{13}C$-CH$_4$ is strong, and poorly constrained by models, to the degree that it prohibits the reliable atmospheric reconstruction of this parameter from firn air measurements (Sapart et al., 2013). Diffusive fractionation is another type of kinetic fractionation and can be tested in our model. We assume a constant atmospheric $^{13}C/^{12}C$ isotope ratio of 1.1147302 % for CO$_2$ ($\delta^{13}C$-CO$_2$ = -8 ‰) and 1.0709052 % for CH$_4$ ($\delta^{13}C$-CH$_4$ = -47 ‰) respectively. Thermal fractionation and gravitational settling are neglected to highlight only the impact of the atmospheric mixing ratio change. The model including barometric pumping calculates $\delta^{13}C$-CO$_2$ and $\delta^{13}C$-CH$_4$ values depleted by up to ~0.2 ‰ and ~1.97 ‰ relative to the atmosphere in the WAIS Divide LIZ at the time of firn air sampling (Fig. 15). Without barometric pumping, delta values are notably higher because molecular diffusion is stronger, and the dispersive mixing no longer smooths out the profile in the deep firn.



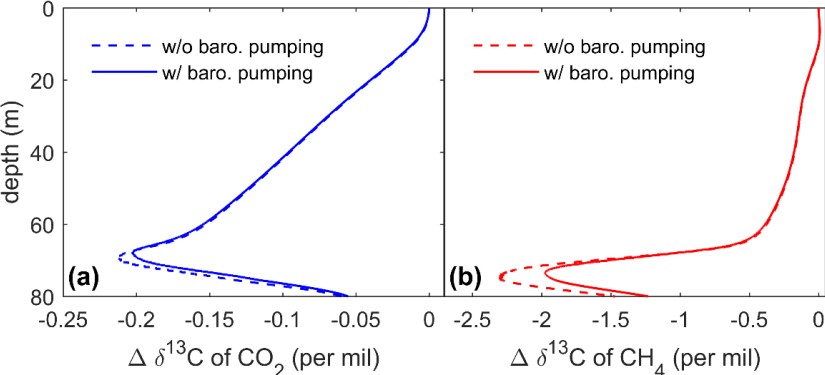

**Figure 15.** Diffusive fractionation effect at the time of sampling at WAIS Divide on $\delta^{13}C$ of (a) $CO_2$ and (b) $CH_4$. Atmospheric mixing ratios of $^{12}CO_2$, $^{13}CO_2$, $^{12}CH_4$ and $^{13}CH_4$ were obtained from atmospheric trace gas histories used to drive the firn air model (SI) and assuming constant atmospheric isotope ratios of -8 ‰ and -47 ‰ for $\delta^{13}C$-$CO_2$ and $\delta^{13}C$-$CH_4$, respectively. Firn air values are presented as the difference from the constant atmospheric isotope ratios.

### 4.3    Predicting disequilibrium

Past mean ocean temperature can be estimated from the noble gas concentrations in ice core bubbles (Headly and Severinghaus, 2007; Ritz et al., 2011). On glacial-interglacial timescales, the concentrations of noble gases are exclusively controlled by dissolution in the ocean. Because the temperature sensitivity of solubility is different for each gas, measurements of noble gas ratios in ice cores can be used to obtain a signal of integrated ocean temperature. However, as for any gas, the trace gas concentrations in bubbles must first be corrected for alterations of the atmospheric signal in the firn.

In an attempt to compensate implicitly for disequilibrium effects and gravitational settling at the same time, it has been suggested that the elemental ratios $Kr/N_2$ and $Xe/N_2$ in bubbles should be corrected by subtracting krypton or xenon isotope ratios, respectively (Headly, 2008). The hope is that krypton and xenon isotopes may be influenced similarly by the processes responsible for creating disequilibrium in $Kr/N_2$ and $Xe/N_2$. Therefore, this approach may compensate for disequilibrium effects and gravitational settling simultaneously, but it has been untested in models so far. $\epsilon'$ values modelled here allow us to evaluate this method quantitatively. We use a linear fit to predict the $\epsilon'_{Kr/N_2}$ from $\epsilon'_{Kr}$. The linear fit yields good agreement with the modelled $\epsilon'_{Kr}$ over the whole firn column (R²>0.998), indicating that the scaling between $\epsilon'$ values is nearly independent of depth. We find that (mass-normalized) $\epsilon'_{Kr/N_2}$ should be approximately 75 % of the (mass-normalized) $\epsilon'_{Kr}$ at WAIS Divide. Scaling relationships for other isotope and element pairs are shown in Table 1 and are equally robust. Moreover, our results show that the source of disequilibrium is irrelevant to the correction for the macroscopic processes represented in our model. Advection and convective or dispersive mixing show the same scaling relationships for $\epsilon'$. At Law Dome DSSW20K, the calculated ratio of $\epsilon'_{Kr/N_2}$ and $\epsilon'_{Kr}$ is at 75.9% almost identical to the result at WAIS Divide. Sensitivity tests with the 1D analytical model presented in Appendix A demonstrate that the disequilibrium scaling relationship between Kr isotopes and $Kr/N_2$ is robust to within $\pm 5\%$ over a wide parameter range of molecular diffusivity, eddy diffusivity and





advection velocity. Uncertainties become largest in the extreme case when $\epsilon'_{Ar}$, the lowest simulated $\epsilon'$ value, is used to predict $\epsilon'_{Xe/Kr}$, the highest simulated $\epsilon'$ value, but never exceed $\pm\ 25\%$.

This suggests that the same scaling relationship between $\epsilon'_{Kr/N_2}$ and $\epsilon'_{Kr}$ may be used at any firn sampling site without introducing large biases. Although predicted $\epsilon'_{Kr}$ values at WAIS Divide are close to the current analytical uncertainty of the
[86]Kr/[82]Kr measurement, correcting for kinetic fractionation and disequilibrium will become advisable with future improvements in precision and may improve mean ocean temperature reconstructions.

**Table 1.** $\epsilon'$ scaling factor in the 2D model with barometric pumping between different element and isotope ratios from linear regression of $\epsilon'$ value pairs at all depths. $R^2 > 0.996$ for all relationships.

| Predictor: isotope ratio | Response: time-variable atmospheric gas ratios | | |
|---|---|---|---|
| | $\epsilon'_{Kr/N_2}$ | $\epsilon'_{Xe/Kr}$ | $\epsilon'_{Xe/N_2}$ |
| $\epsilon'_{Ar}$ | 3.94 | 8.74 | 6.16 |
| $\epsilon'_{Kr}$ | 0.75 | 1.66 | 1.17 |
| $\epsilon'_{Xe}$ | 0.54 | 1.19 | 0.84 |

## 5 Conclusions

We developed a two-dimensional firn air transport model that explicitly represents tortuosity in the firn column through migrating layers of reduced permeability. The idealized representation of firn layering is physically motivated and may illustrate the impact of firn density anomalies (i.e., summer vs winter firn or wind crusts) on gas transport. The model also accounts for thermal fractionation, a convective zone, and surface pressure-forced barometric pumping. Dispersive mixing resulting from barometric pumping is constrained in the model by previously published parametrizations and not subject to
tuning. Simulations of the $\delta^{15}N$ profile at WAIS Divide show that extensive horizontal diffusion through the tortuous firn structure is required by the discontinuous layers. This limits the effective vertical diffusion of gases at depth. However, layering alone cannot prevent gravitational enrichment of isotopes in the deep firn completely. Similarly, the effect of barometric pumping alone is insufficient to obtain agreement with observations. The combination of barometric pumping with layering in contrast leads to amplified dispersive mixing by velocity focusing in layer openings and a more natural emergence of a lock-
in zone in the model.

Downward advection, convective mixing and dispersive mixing all hinder trace gases in reaching the isotope ratios expected from gravitational settling. This kinetic fractionation is strongest for slow diffusing gases and increases with firn column height. Our experiments show clearly that barometric pumping leads to increased isotopic disequilibrium in the firn column. However, our experiments fail to account for the large [86]Kr excess observed in the WAIS Divide core, as well as for
the relatively weak $\delta^{15}N$ enrichment seen at DSSW20K, suggesting that these effects are not caused by the presence of layering (as previously suggested) and that their origin must be sought elsewhere. We further find robust scaling relationships between the magnitude of disequilibrium in different noble gas isotope and elemental ratios. Our results suggest that, to first order,





these relationships are independent of depth in the firn column and independent of the reason for disequilibrium for the process represented in the model (i.e., dispersive mixing, advection or convective mixing). Thus, a correction may be applied to measured noble gas ratios in the reconstruction of mean ocean temperature to account for kinetic fractionation.

## 6    Appendices

### 6.1    Appendix A: An analytical solution for simplified firn air transport

Here, we seek an analytical solution to the following idealized scenario of firn air transport: firn air advection, diffusion, and dispersion in one dimension. In this case, vertical trace gas migration relative to the major gas nitrogen is governed by the Eq. (A1):

$$\tilde{s}\frac{\partial q}{\partial t} = \frac{\partial}{\partial z}\left[\tilde{s}D_m\left(\frac{\partial q}{\partial z} - \frac{\Delta m\, g}{R\, T}q + \Omega\frac{\partial T}{\partial z}q\right) + \tilde{s}D_e\frac{\partial q}{\partial z}\right] - \tilde{s}w\frac{\partial q(z,t)}{\partial z}, \tag{A1}$$

with $q \equiv \delta + 1$ the ratio of an isotope to $^{28}N_2$ relative to a standard, $\tilde{s} \equiv s_{op}\exp\left(\frac{\Delta mgz}{RT}\right)$ the pressure-corrected open porosity ($m^3\ m^{-3}$), $D_m$ and $D_e$ the molecular and eddy diffusivity ($m^2\ s^{-1}$), $\Omega$ the thermal diffusion sensitivity ($K^{-1}$), and $w$ the effective air advection velocity due to snow accumulation and pore compression ($m\ s^{-1}$) (e.g., Schwander et al., 1993; Rommelaere et al., 1997; Trudinger et al., 1997; Severinghaus et al., 2010; Buizert et al., 2012; Kawamura et al., 2013). The five terms on the right-hand side of Eq. (A1) represent Fickian diffusion, gravitational settling, thermal fractionation, mass-independent dispersion and gas advection (from left to right). A Dirichlet (i.e., known value) boundary condition is chosen at the top of the firn column and represents the well-mixed atmosphere. The bottom boundary condition is given by a Neumann boundary condition allowing only an advective flux to leave the domain.

Assuming steady-state and neglecting changes of $\tilde{s}$, $D_m$, $D_e$ and $w$ with depth, Eq. (A1) reduces to (Severinghaus et al., 2010)

$$\frac{\partial q}{\partial t} \equiv 0 = (D_m + D_e)\frac{\partial^2 q}{\partial z^2} - D_m(G - \mathcal{T})\frac{\partial q}{\partial z} - w\frac{\partial q}{\partial z}, \tag{A2}$$

where $G \equiv \frac{\Delta m\, g}{R\, T}$ and $\mathcal{T} \equiv \Omega\frac{\partial T}{\partial z}$ represent the constants in the gravitational and thermal fractionation term.

The solution for trace gas profiles in delta notation takes the form

$$\delta = \begin{cases} \dfrac{\exp\left(\dfrac{D_m(G - \mathcal{T}) + w}{D_m + D_e}\, z\right) - 1}{\dfrac{w}{D_m(G - \mathcal{T})}\exp\left(\dfrac{D_m(G - \mathcal{T}) + w}{D_m + D_e}\, z_{COD}\right) + 1}, & z \leq z_{COD}, \\[4ex] \delta(z_{COD}), & z > z_{COD} \end{cases} \tag{A3}$$





where $z_{COD} = z(COD)$ (see SI Fig. S2). Note that Eq. (A2) only applies to trace gas transport into N$_2$, not to transport of one trace gas into another trace gas, as discussed in the text. Nevertheless, the equation can be used as such to calculate $\delta^{15}$N.

By evaluating some extreme cases, Eq. (A4) illustrates a few key points about trace gas transport of $\delta^{15}$N in firn. Under a large negative temperature gradient (i.e., atmospheric warming, $\mathcal{T} \to -\infty$), $\delta \to \infty$ and thermally sensitive gases are enriched in the firn because the numerator grows faster than the denominator. Similarly, heavier gases ($G \to \infty$) are more strongly fractionated ($\delta \to \infty$) than lighter gases assuming they have the same molecular diffusivity. Advection ($w \to \infty$) and eddy mixing ($D_e \to \infty$) prevent the system from reaching the trace gas concentrations expected from gravitational settling and ultimately force concentrations to be constant ($\delta \to 0$). A lack of molecular diffusion ($D_m \to 0$) leads to the same result ($\delta \to 0$). Naturally, Eq. (A4) reduces to the profile of a gravitationally settled gas (i.e., Eq. (1)) when $w \to 0$ and $D_e \to 0$.

## 6.2    Appendix B: Differential kinetic isotope fractionation in ratios of two different elements

Here we revisit the relative disequilibrium for ratios of two elements as seen in Figs. 12 and 13. First, recall the definition of $\epsilon'$ for a ratio of isotopes $x$ and $y$ (indicated by their nominal atomic masses)

$$\epsilon'_{x/y} \equiv \ln\left(\frac{q_{x/y}^{\frac{1}{m_x - m_y}}}{q_{29/28}}\right). \tag{A4}$$

Equation (3) shows that $q_{x/y}$ is the ratio of $q$ values calculated for the transport of each isotope into $^{28}$N$_2$ ($q_{x/28}$ and $q_{y/28}$). $q_{x/28}$ (or $q_{y/28}$) may also be expressed in reference to nitrogen using the $\epsilon'$ value for the isotope

$$\ln\left(q_{x/28}^{\frac{1}{m_x - m_{28}}}\right) = \ln(q_{29/28}) + \epsilon'_{x/28} = \ln\left(q_{29/28} \cdot \exp(\epsilon'_{x/28})\right). \tag{A5}$$

Note that $\epsilon'$ by definition is already mass-normalized. It follows from Eqs. (3), (A4) and (A5) that

$$\epsilon'_{x/y} = \ln\left(\frac{\left(\frac{q_{x/28}}{q_{y/28}}\right)^{\frac{1}{m_x - m_y}}}{q_{29/28}}\right) = \ln\left(\frac{\left(\frac{\left(q_{29/28} \cdot \exp(\epsilon'_{x/28})\right)^{m_x - m_{28}}}{\left(q_{29/28} \cdot \exp(\epsilon'_{y/28})\right)^{m_y - m_{28}}}\right)^{\frac{1}{m_x - m_y}}}{q_{29/28}}\right). \tag{A6}$$

Equation (A6) may be rewritten to yield

$$\epsilon'_{x/y} = \frac{m_x - m_{28}}{m_x - m_y}\left[\ln(q_{29/28}) + \epsilon'_{x/28}\right] - \frac{m_y - m_{28}}{m_x - m_y}\left[\ln(q_{29/28}) + \epsilon'_{x/28}\right] - \ln(q_{29/28}). \tag{A7}$$





Because the terms containing $q_{29/28}$ cancel, we obtain a straightforward expression to find $\epsilon'$ for any isotope ratio from the $\epsilon'$ of two nuclides relative to $^{28}N_2$

$$\epsilon'_{x/y} = \frac{m_x - m_{28}}{m_x - m_y}\epsilon'_{x/28} - \frac{m_y - m_{28}}{m_x - m_y}\epsilon'_{y/28} \,. \tag{A8}$$

Analysis of this relationship reveals that disequilibrium should most strongly affect ratios of two heavy isotopes, such as $^{132}Xe/^{84}Kr$, because heavy elements diffuse slower than $N_2$ (i.e., $\epsilon'_{x/28} \ll 0$) and the mass weighting factor larger in the first than in the second term (i.e., $\frac{m_x - m_{28}}{m_x - m_y} \gg \frac{m_y - m_{28}}{m_x - m_y}$). As long as no explicit correction for disequilibrium effects is implemented in the determination of mean ocean temperature, this implies that particular caution should be used in interpreting the $^{132}Xe/^{84}Kr$ record at sites with substantial disequilibrium.

Although this equation can theoretically predict $\epsilon'$ of any isotope ratio from $\epsilon'$ of the two isotopes $x$ and $y$ relative to $^{28}N_2$ (i.e., $\epsilon'_{x/28}$ and $\epsilon'_{y/28}$), in practice, this approach will not allow correcting for differential kinetic isotope fractionation. $\epsilon'_{x/28}$ cannot be measured directly and the atmospheric ratio of the noble gas $x$ to nitrogen is not constant over long timescales. Thus, $\epsilon'_{x/28}$ will not only be affected by disequilibrium but will also be influenced by atmospheric variability resulting from gas specific solubility differences (i.e., precisely the mean ocean temperature signals we attempt to reconstruct). Instead we suggest that the scaling relationships provided in Sect. 5.3 can be used to predict the $\epsilon'$ of noble gas elemental ratios.

## 7 Competing interests

The authors declare that they have no conflict of interest.

## 8 Acknowledgements

We would like to thank Jakob Keck, Alan Seltzer and Ian Eisenman for providing computational resources and insightful discussions on the numerical implementation of firn air transport. Sarah Shackleton has provided helpful comments on the importance of disequilibrium in mean ocean temperature reconstruction. This work was supported by NSF grant PLR-1543229 and PLR-1543267.

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
