# Peer review of "The influence of layering and barometric pumping on firn air transport in a 2D model"

_The Cryosphere, 2017_

## Referee Comment (RC1) · S. Drake (Referee) · 13 Jan 2018

Journal: The Cryosphere

Manuscript: The influence of layering and barometric pumping on firn air transport in a 2D model

Authors: Benjamin Birner, Christo Buizert, Till J.W. Wagner and Jeffery P. Severinghaus

MS No.: tc-2017-233

MS Type: Research article

Reviewer: Stephen Drake

[Figure]

Date: Friday, January 12, 2018

Overview

This manuscript addresses firn air mixing stimulated by barometric pumping. The authors have developed a 2D model that simulates advection, convection, dispersion, and diffusion of air through firn that has discontinuous, low-permeability layers. They apply this model to investigate the relative impacts of diffusion and dispersion with depth for several noble gas isotopologues. Improved understanding of firn air mixing will enable more accurate assessments of ancient atmospheric composition and climate change chronology derived from firn and ice cores.

I recommend this manuscript for publication once the following items have been addressed.

General Comments

A result of Buizert and Severinghaus (2016) is that pressure changes above the snowpack manifest with full amplitude down to the lock-in zone after $\sim$ 1 hr. This one hour timescale means that mesoscale and diurnal pressure variations may also influence firn air mixing. Synoptic pressure changes have more spectral power than mesoscale and diurnal pressure changes, however, mesoscale and diurnal pressure changes are more frequent. So, there is an interplay between frequency and amplitude that was not addressed in either this manuscript or in Buizert and Severinghaus (2016). Also, it is relevant to note that synoptic pressure changes do not always yield storms. These complicating factors perhaps would have been better addressed in Buizert and Severinghaus (2016). I leave it to the editor to decide whether they should be addressed in this (already substantial) manuscript.

The authors did not attempt to assess the error that could be attributed to barometric pumping when dating the composition of the atmosphere using ice cores. It would be instructive to a broader audience if they speculate as to what other information is

needed to bound this problem.

Clarity

Overuse of "here" especially to begin a sentence.

It would be helpful if, rather than intermixing "impermeable" and "(near-) impermeable", the authors choose one term, define its meaning in the introduction and use it throughout the text (with the exception of the references to model layers, which are explicitly and accurately defined as impermeable).

Throughout most of the manuscript, the vertical dimension is referred to as "depth". But on pages 14 and 15 it is referred to as "height". I prefer that they use "depth" throughout.

The authors and others have demonstrated that interstitial air mixing near the snow surface is driven both convectively (by temperature gradients) and dynamically (through pressure changes). This is an opportunity to rename the upper layer as a "mixed" zone (or some-such) rather than "convective" zone so as not to perpetuate the overly-simplistic convective zone terminology.

It is easier to parse the history of scientific discovery when multiple references are listed chronologically in the manuscript.

Specific Comments

Line 9: "impermeable" do you mean low permeability?

Line 16: "Moreover, we find that . . ." This is a confusing sentence, consider re-wording. Do you mean: As observed in nature, simulated barometric pumping does not substantially change the differential fractionation of fast and slow moving gases?

Line 18: "This suggests that . . ." what is "This" ?

Figure 1: consider brightening/enhancing this image so the layers are easier to distinguish

Line 11: "... smoothing out any concentration gradients ..." As shown in Drake et al. (The Cryosphere, 2017) snow inhomogeneities provide preferred pathways for airflow. So, remove the "out any".

Line 13: "Such convective mixing ... " The authors are convolving convection with a non-convective pressure-driven process.

Fig 2a: This idealized medium is not non-dispersive. The streamlines, as drawn, are not realistic. Airflow around one sphere has close to an equal chance of going around the next sphere on either side with the net effect that particles will spread out both in the streamwise and transverse directions.

Line 3: Why do the pressure-induced air flows need to be fast? Are you suggesting turbulent mixing? How fast is fast?

Line 5: "... emergent macroscale phenomenon ..." what is emergent about dispersion?

Line 7: is dispersion added to the governing equation or is it rather not removed from a simplified form of the governing equation?

Line 18: "... hindering effect..." how about "diminishing effects"?

Line 10: "Most current 1D firn air models..." does Buizert et al. (2012) contain a review of other 1D firn air models?

Line 17: "... discontinuous layers of zero diffusivity and barometric pumping" how about, "...by barometric pumping and discontinuous layers that have nominal diffusivity"?

Line 21: "the driving force for gravitational settling is effectively zero during horizontal transport ..." need to reword because the gravitational force is not zero regardless of horizontal transport

It would be more complete to also define the term on the LHS of Eq (2).

Fig 4c: Why is the maximum in barometric pumping at $\sim$ 15m depth?

Line 5: "... assuming a constant snow and ice mass flux at all depths" How does this assumption bias your results (if at all)?

Line 13: "... barometric pumping in the more tortuous, deep firn" barometric pumping occurs throughout the firn column, not just in deep firn

Line 3: "... longitudinal to-flow and transverse to-flow" how about "streamwise and cross-stream"?

Line 5: Is there a quantifiable basis for the assumption of 10x difference between horizontal and vertical molecular diffusivities?

Fig 7. Comment on two $CO_2$ (diamond) anomalies as it appears they are ignored in

[Figure]

the curve fits.

Line 2: "In line with observations, . . ." the authors previously indicated an anthropogenic signal in CO2 and CH4 that could be repeated here for clarity

Fig. 8: what do error bars with missing end caps mean relative to the error bars that have end caps?

Line 20: "A lack of alternative pathways" could be stated as "Fewer alternative pathways" or "Limited number of alternative pathways" or similar

Line 32: If thermal effects are neglected then why is it called the convective zone? Perhaps rephrase sentences in lines 31 and 32.

Line 11: "Constraining the convective zone . . . much larger than at the more recently sampled WAIS site." need citations

Fig 11: Do the 2D simulations include impermeable layers? What is the meaning of missing error bar caps?

Line 9: "Advection and mass-independent mixing. . ." is this the authors' theory or is a citation needed?

Line 7: "Therefore, ratios of heavier elements are more susceptible to kinetic fractionation." This sentence needs unpacking/rephrasing.

Eq 13: Since this is not strictly a Péclet number you could reference it as a modified (enhanced?, dispersive?) Péclet number.

Line 20: how are these values chosen?

Lines 21-22: " Downward advection ..." It should be clarified that these two sentences are based on previously reported results rather than new insights derived from this investigation. For example, "Previous studies (citations) have shown that ..."

Line 2: explicitly specify the correction or explain the correction in more detail

Line 20: source for the solution found in Eq (A3) to Eq (A2)?

Line 2: "Nevertheless ..." Need citation or rationale for why Eq (A2) can be used to calculate $\delta^{15}N$

Technical Corrections

Line 14: "supresses" → suppresses

Line 2: " ... unconsolidated snow ..." I think of firn as consolidated snow.

Line 10: "processes which" → processes, which

Line 14: "high resolution" → high-resolution

Line 15: remove "the same amount of"

Line 1: "Last" → Lastly

Line 3: "gradients and induce" → gradients that induce

Line 17: "DZ but the effective" → DZ but effective

Line 7: change " becomes" to "simplifies to" (even though you already have "simplifies to" in the text just above)

Line 6: "Kawamura (unpublished)"  - might as well leave this out since you have another citation

Line 4: "In the following we" → In the following discussion we

Line 9: "Isotopes ratios are higher" → Simulated isotope ratios are higher

Line 1: "the dispersivity" → dispersivity

Line 4: "estimate is within" → estimate of 2.8 m is within

Line 17: "... 2013)as" → ... 2013) as

[Figure]

Line 13: "the isotopologues" → remove the "the"

Line 20: "at WAIS Divide" → at the WAIS Divide

Line 23: "75.9% almost" → 75.9%, almost

Line 22: remove the "This"

Line 23: remove "clearly"

Line 1: "these relationships" → these scaling relationships

Line 10: "Movment" → Movement

Line 11: "seperation" → separation

Technical Corrections for Supplement

Line 2: "q is a Nx1 vectors" → q is an Nx1 vector

Line 5: "off diagonals" → off-diagonals

Line 1: "at all depth" → at all depths

Fig S3: Does this figure indicate that there is a $\sim$ 35 year delay in the response of the

$\delta\hat{}15$ N profile due to the 3-day time step relative to the 3 day/5 $\sim$ 14 hr time step?

Line 3: "Eularian" → Eulerian

Fig S6: "Plot is shown at reduced grid resolution for clarity." Is this plot in reduced grid resolution or is it a subset of the domain at the original resolution (or both)?

Note: there is a gap in the text partway down the page

Line 4: "firnf"?

Line 6: is there a reference for equation S28?

Line 1: "serval harmonic"?

Table S1: for the WAIS Divide the daily pressure change is 5hPa. Does this mean that for a 3-day time step the pressure change is 15hPa? Or is the 3-day pressure change no more than 15hPa and is quasi-randomly attenuated to match a red-shifted spectra? Or ?

---

## Referee Comment (RC2) · Anonymous Referee #2 · 15 Jan 2018

**The influence of layering and barometric pumping on firn air transport in a 2D model**

Benjamin Birner et al.

General comments: The submitted manuscript analyses the influence of impermeable layer and barometric pumping (driven by surface pressure variability) on firn air transport using a 2D trace gas advection-diffusion-dispersion model accounting for discontinuous horizontal layers of reduced permeability. The simulated results are compared with field measurements from WAIS Divide and Law Dome DSSW20K and show good agreements.

Specific comments: The manuscript is well written and shows interesting results which are sufficient to be published. However, I would suggest to add some minor comments to make the manuscript clearer:

1. In the Figures, I would mention the time step you used for the plots. Are they consistent with the field measurements?
2. In the 'supplementary information' you show a table with some simulation parameters, however some information is missing or is unclear, respectively. (e.g. simulation time, time range of the simulations, resolutions, etc.)
3. I would recommend to also show plots of the boundary conditions at the surface, like temperature, pressure, etc.

Detailed comments manuscript:

Page 3, Line 5 – 6: I would recommend to make 'z', 'T', and 'R' italic to make it consistent.

Page 7, Line 2 – 3: Have you checked the influence of airflow due to temperature changes, especially in the convective zone? Normally, in the top 10-15 meters you have a high variation in the temperature profile due to the changing temperature at the top. This variation in the temperature profile can cause an airflow.

Page 11, Line 4: '… assumed to be temperature independent if the temperature sensitivity is unknown.' -> Where does this temperature sensitivity appear?

Page 11, Figure 7: There are two outliers in the observed $CO_2$ concentrations at around 15 and 50 meters. Is there a reason for this?

Page 11, Figure 7 caption: You refer to Fig. S9 to illustrate the differences in the $CO_2$ and $CH_4$ profiles between the 1D and the 2D model with or without barometric pumping which is not visible at the resolution of this Figure 7. However, Fig. S9 shows the same figure like Fig. 6 but only for Law Dome DSSW20K. Where can I find the Figure of the differences in the $CO_2$ and $CH_4$ profiles between the 1D and the 2D model with or without barometric pumping?

Page 12, Line 10: See my comment to Page 11, Figure 7 caption.

Page 17, Line 13 – 14: What could be the reason that the simulated $^{86}$Kr excess is significant lower than the observed one?

Page 18, Line 6 – 7: 'Heavy, slow-diffusing isotopes approach gravitational equilibrium more slowly than lighter, faster-diffusing isotopes.' -> Maybe it is a silly question but is it not the opposite around? If the isotopes are heavy you have a faster settlement due to stronger gravitational force?

Page 22, Line 24: 'However, our experiments fail to …' -> Which experiments? I cannot see in your manuscript that you did experiments, just simulations.

Page 24, Line 3 and 9: I think you want to refer to Eq. (A3) instead to Eq. (A4).

Detailed comments supplementary information:

Page 2, Equation S5: '$q$' on the left-hand side of the equation is missing.

Page 4, Line 15 – 20: Does it mean that snow accumulation is included?

Page 10, Line 4: How long does it take to run one simulation with time steps of around 3 days?

Page 11, Section 'Thermal model': Can you show a figure of the temperature profile to get an impression of the boundary condition?

Page 11/12: The line break is wrong.

Page 12, Line 4: The line break is wrong and it should mean 'firn'

Page 12, Line 10: Please show a figure of the surface temperature histories.

Page 13, Table S1: Are the parameters the same for WAIS Divide and Law Dome DSSW20K if you only show one number, e.g. the 'Horizontal' or 'Vertical grid spacing'?

Page 13, Table S1, Row 'Width': Does this mean there is a variation of the width? What are the exact values for WAIS Divide and Law Dome DSSW20K you used in the simulations?

Page 13, Table S1, Row 'Depth of first layer': Can you provide an exact value?

Page 13, Table S1, Row 'Temperature': Did you use a fix temperature value at the top? No daily or seasonal variations?

Page 13, Table S1, Row 'Surface Pressure: Did you use a fix pressure value at the top? No daily or seasonal variations?

Page 13, Table S1, Row 'Free air relative diffusivities to $CO_2$': Can you show the value you cited from the Paper?

Page 15, Line 5: '… using q-values.' -> please change it to '*q*-values'.

---

## Author Comment (AC1) · 13 Apr 2018

San Diego, April 12, 2018

**Author response to the reviewers' comments on "The influence of layering and barometric pumping on firn air transport in a 2D model"**

We would like to thank Stephen Drake and an anonymous reviewer for their detailed and insightful comments on our manuscript. Reviewer comments are shown in red below, with our responses in black. We have revised the manuscript following their advice and present a marked-up version of the manuscript and supplementary information at the end of this document.

Best regards,
Benjamin Birner, Christo Buizert, Till Wagner and Jeff Severinghaus

**Changes to the model**
A minor numerical mistake in the spacing of layers (layers were separated by slightly more than 1 annual layer thickness) was corrected. The amplitude of surface pressure variability in the model was changed to accurately reflect the time step dependence of the pressure forcing and the attenuation of the amplitude f barometric pumping in the model was removed. These corrections mostly compensate each other.
The modifications implemented did not change any of our results appreciably, but all figures were updated in the text.

**First reviewer, Stephen Drake**

**Overview**
This manuscript addresses firn air mixing stimulated by barometric pumping. The authors have developed a 2D model that simulates advection, convection, dispersion, and diffusion of air through firn that has discontinuous, low-permeability layers. They apply this model to investigate the relative impacts of diffusion and dispersion with depth for several noble gas isotopologues. Improved understanding of firn air mixing will enable more accurate assessments of ancient atmospheric composition and climate change chronology derived from firn and ice cores.
I recommend this manuscript for publication once the following items have been addressed.

**General Comments**
A result of Buizert and Severinghaus (2016) is that pressure changes above the snowpack manifest with full amplitude down to the lock-in zone after ~ 1 hr. This one hour timescale means that mesoscale and diurnal pressure variations may also influence firn air mixing. Synoptic pressure changes have more spectral power than mesoscale and diurnal pressure changes, however, mesoscale and diurnal pressure changes are more frequent. So, there is an interplay between frequency and amplitude that was not addressed in either this manuscript or in Buizert and Severinghaus (2016). Also, it is relevant to note that synoptic pressure changes do not always yield storms. These complicating factors perhaps would have been better addressed in Buizert and Severinghaus (2016). I leave it to the editor to decide whether they should be addressed in this (already substantial) manuscript.

We agree with the reviewer that there is a complex interplay between frequency and amplitude that controls barometric pumping in firn. The rate of change in surface pressure determines the airflow velocity in firn and therefore the resulting dispersive mixing. Fast flows can be induced by rapid pressure variations of small amplitude or by slower changes with larger amplitude. In addition, the frequency-dependent attenuation of pressure waves with depth further complicate the situation. This is illustrated by the surface mixed zone (also called the "convective zone") where pressure variability on short time scales makes an appreciable contribution to mixing and reduces gravitational fractionation but the vertical extent of this region is limited. Unfortunately, our model cannot resolve pressure variability on such short time scales due to computational constraints. Thus, we believe that the role of mesoscale and diurnal pressure changes is beyond the scope of this study and would better be addressed in separate publication. Note that Buizert and Severinghaus (2016) in fact used 6h resolution data surface pressure forcing for their 1D model, so that in principle diurnal pressure variability should have been accounted for.

The authors did not attempt to assess the error that could be attributed to barometric pumping when dating the composition of the atmosphere using ice cores. It would be instructive to a broader audience if they speculate as to what other information is needed to bound this problem.

We thank the reviewer for raising this point. Barometric pumping typically causes deviations from gravitational equilibrium that are orders of magnitude smaller than the analytical precision of $CO_2$ and $CH_4$ measurements. We therefore assume that the effect of barometric pumping on these records may safely be neglected and is not discussed in the text. Barometric pumping could reduce $\delta^{15}N$ below the gravitational equilibrium by a small amount; $\delta^{15}N$ is commonly used to infer gas age-ice age differences. In that sense it may influence the dating. However, in most sites this $\delta^{15}N$ effect is very small, certainly much smaller than other uncertainties such as the past thickness of the surface mixed zone.

**Clarity**

Overuse of "here" especially to begin a sentence.

Thank you. The use of "here" was considerably reduced.

It would be helpful if, rather than intermixing "impermeable" and "(near-) impermeable", the authors choose one term, define its meaning in the introduction and use it throughout the text (with the exception of the references to model layers, which are explicitly and accurately defined as impermeable).

We now use the terminology "reduced permeability" to describe the influence of layers in nature and reserve the term "impermeable" for layers in the model realm. Details on the numerical implementation of layers may be found in the Supplementary Information (SI).

Throughout most of the manuscript, the vertical dimension is referred to as "depth". But on pages 14 and 15 it is referred to as "height". I prefer that they use "depth" throughout.

"Height" was replaced by "depth".

The authors and others have demonstrated that interstitial air mixing near the snow surface is driven both convectively (by temperature gradients) and dynamically (through pressure changes). This is an opportunity to rename the upper layer as a "mixed" zone (or some-such) rather than "convective" zone so as not to perpetuate the overly simplistic convective zone terminology.

We thank the reviewer for this comment and fully agree. We renamed the "convective zone" to "surface mixed zone" throughout the manuscript to highlight this important dual nature of mixing in the near-surface region. A brief rational for this change was added to the introduction.

It is easier to parse the history of scientific discovery when multiple references are listed chronologically in the manuscript.

References are now in chronological order. Reordering of references is not highlighted in the attached marked-up version of the document to maintain easy readability.

**Specific Comments**

Line 9: "impermeable" do you mean low permeability?
Yes (manuscript corrected).

Line 16: "Moreover, we find that…" This is a confusing sentence, consider re-wording. Do you mean: As observed in nature, simulated barometric pumping does not substantially change the differential fractionation of fast and slow moving gases?
There is little impact of barometric pumping on the differential fractionation of fast and slow-moving gases in our model. However, this modelling result does not agree with some preliminary observations from 5 different Antarctic ice coring sites which indicate that there is a coherent correlation between the amplitude of pressure variability at a site and the measured krypton excess (Buizert, unpublished). The sentence was restructured to better reflect this connection.

Line 18: "This suggests that…" what is "This" ?
We mean the shortcoming of the model described in the previous sentence (see comment above).

Figure 1: consider brightening/enhancing this image so the layers are easier to distinguish
The contrast of the image was improved.

Line 11: "…smoothing out any concentration gradients…" As shown in Drake et al. (The Cryosphere, 2017) snow inhomogeneities provide preferred pathways for airflow. So, remove the "out any".
Done

Line 13: "Such convective mixing…" The authors are convolving convection with a non-convective pressure-driven process.
The sentence was changed to better reflect the distinction between the two processes.

Fig 2a: This idealized medium is not non-dispersive. The streamlines, as drawn, are not realistic. Airflow around one sphere has close to an equal chance of going around the next sphere on either side with the net effect that particles will spread out both in the streamwise and transverse directions.
That is correct. We have decided to remove the figure from the manuscript since the figure was previously published and we would like to shorten the overall length of the paper

Line 3: Why do the pressure-induced air flows need to be fast? Are you suggesting turbulent mixing? How fast is fast?
Buizert and Severinghaus (2016) have demonstrated the fast propagation of pressure waves in firn returning the column to hydrostatic balance. These readjustment flows are fast compared to other air flows in firn (see Fig. 4 in the text) but generally $<10^{-4}$ m/s. Therefore, we do not expect to see turbulent mixing in the porous firn medium as the Reynolds number of the flow is very small ($Re<<1$).

Line 5: "…emergent macroscale phenomenon…" what is emergent about dispersion?

We appreciate that this was not fully clear. We are referring to dispersion emerging when you have interactions of the fluid with the porous medium. On scales smaller than the pore-scale, only advection and diffusion control tracer distributions.

Line 7: is dispersion added to the governing equation or is it rather not removed from a simplified form of the governing equation?
The full governing equation indeed should contain dispersion but traditionally this term has been neglected in firn. The text was changed accordingly.

Line 18: "…hindering effect…" how about "diminishing effects"?
Manuscript corrected.

Line 10: "Most current 1D firn air models…" does Buizert et al. (2012) contain a review of other 1D firn air models?
Yes, Buizert et al. (2012) is a model intercomparison study and discusses different implementations of the lock-in zone in a range of firn air models.

Line 17: "…discontinuous layers of zero diffusivity and barometric pumping" how about, "…by barometric pumping and discontinuous layers that have nominal diffusivity"?
Thank you, we followed the reviewer's recommendation.

Line 21: "the driving force for gravitational settling is effectively zero during horizontal transport…" need to reword because the gravitational force is not zero regardless of horizontal transport.
We modified the sentence to "[…] vertical settling of isotopes is greatly reduced during horizontal transport along layers […]".

It would be more complete to also define the term on the LHS of Eq (2).
The LHS terms is now included.

Fig 4c: Why is the maximum in barometric pumping at ~15m depth?
The amount of air displaced by barometric pumping decreases monotonically with depth in the firn column. However, if we scale velocity by the size of the pores (as shown in the figure), velocity becomes a function of volume transport and cross-sectional area (i.e., open porosity). The velocity per area is largest around 15 m depth because the firn is considerably more porous above and less total volume transport occurs in the regions below. Slower velocities in the top few meters of the firn correspond to more total volume transport but the transport is offset by higher porosity. We plot the velocity per area pore space as this velocity is responsible for causing dispersion.

Line 5: "…assuming a constant snow and ice mass flux at all depths" How does this assumption bias your results (if at all)?
We assume that firn properties such as density, open porosity and accumulation are constant in time. Furthermore, we assume conservation of mass when snow accumulates. Therefore, a constant snow/ice mass flux and increasing firn density with depth implies decreases in the downward advection of firn ($w_{firn}$). This is a very common assumption in the firn air transport modelling community, and generally insufficient data on past conditions are available to deviate from it.

We expect that changes in accumulation and firn density on different time scales would modify the porosity and permeability of firn and change the vertical distance of (annual) layers. In fact, the difference in density between summer and winter ice is one observation that motivated the present study of layering in firn.

Line 13: "…barometric pumping in the more tortuous, deep firn" barometric pumping occurs throughout the firn column, not just in deep firn
Correct. What we mean is that barometric pumping induces airflows throughout the firn column but the impact of barometric pumping on trace gas concertation profiles is most prominent near the surface and in the deep firn. Barometric pumping leads to substantial air exchange between the near-surface firn and the unfractionated atmosphere above. Additionally, in the deep firn, the dispersivity of the firn medium is considerably higher than in the shallower sections of the column, therefore barometric pumping yields the strong dispersive mixing in this region despite slower flow velocities. The sentence was changed to reflect this.

Line 3: "…longitudinal to-flow and transverse to-flow" how about "streamwise and cross-stream"?
Based on our reading of the literature (e.g., Freeze and Cherry, 1979) "longitudinal" and "transverse to-flow" are the commonly used direction terms in the context of dispersion. We have decided to follow this convention.

Line 5: Is there a quantifiable basis for the assumption of 10x difference between horizontal and vertical molecular diffusivities?
The reviewer raises an important point. The molecular diffusivity is determined by the tortuosity of firn at the subgrid scale and therefore obtained by tuning. Although it is clear that firn pores are better connected in the horizontal than vertical direction, selecting a factor of 10x difference between horizontal and vertical molecular diffusivities is a somewhat arbitrary choice.

As a sensitivity test, we repeated the numerical experiments with a modified version of the model in which we set this factor to 1x and reduced the horizontal width of impermeable layers by a factor of $\sqrt{10}\sim3.2$ to compensate for the reduction in effective vertical transport caused by changing horizontal diffusivity. This yields qualitatively similar results to previous model runs (Fig. RW1). The observed strength of barometric pumping is slightly reduced in these model runs because less air is forced to flow through the horizontal layers in response to surface pressure changes (barometric pumping flows through layers effectively scale with the area-integrated density change in the regions of the model below a layer).

[Figure]

**Figure RW1**. $\delta^{15}N$ profiles simulated by the model with and without barometric pumping and different choices of horizontal diffusivity ($D_h$). The vertical diffusivities ($D_v$) remain identical, but the horizontal length of layers was scaled by a factor of $\sqrt{10}{\sim}3.2$ to compensate for differences in horizontal diffusivity.

Fig 7. Comment on two CO2 (diamond) anomalies as it appears they are ignored in the curve fits.
Two samples at ~15 m and ~50 m depth were presumably contaminated with modern air based on high $CO_2$ and $CH_4$ and thus ignored in the curve fit. We have added a comment to the respective figure captions.

Line 2: "In line with observations,…" the authors previously indicated an anthropogenic signal in CO2 and CH4 that could be repeated here for clarity
Thank you for this suggestion. The observed and simulated profiles are now explained in a little more detail.

Fig. 8: what do error bars with missing end caps mean relative to the error bars that have end caps?
This was simply a figure resolution problem in the pre-production document. All error bars should have end caps in all post-production, high-resolution images.

Line 20: "A lack of alternative pathways" could be stated as "Fewer alternative pathways" or "Limited number of alternative pathways" or similar
The sentence was rephrased.

Line 32: If thermal effects are neglected then why is it called the convective zone? Perhaps rephrase sentences in lines 31 and 32.
The "convective zone" was renamed to "surface mixed zone" throughout the document.

Line 11: "Constraining the convective zone… much larger than at the more recently sampled WAIS site." need citations

Citation for data added.

Fig 11: Do the 2D simulations include impermeable layers? What is the meaning of missing error bar caps?
Both plotted simulations include impermeable layers. This has been clarified in the caption. Missing end caps on error bars were again an image resolution problem.

Line 9: "Advection and mass-independent mixing…" is this the authors' theory or is a citation needed?
Citation added

Line 7: "Therefore, ratios of heavier elements are more susceptible to kinetic fractionation." This sentence needs unpacking/rephrasing.
The sentence was rephrased.

Eq 13: Since this is not strictly a Péclet number you could reference it as a modified (enhanced?, dispersive?) Péclet number.
We are now using the terminology of a "modified Péclet number" throughout the manuscript.

Line 20: how are these values chosen?
These values approximately represent the present-day atmosphere. Since we report $\delta^{13}C$ values in the firn relative to the atmosphere, the choice of any constant atmospheric ratio does not influence our results.

Lines 21-22: "Downward advection…" It should be clarified that these two sentences are based on previously reported results rather than new insights derived from this investigation. For example, "Previous studies (citations) have shown that…"
Citations were added.

**Line 2: explicitly specify the correction or explain the correction in more detail**
The correction is outlined in more detail now and the total magnitude of the effect estimated following previously published methods (Bereiter et al., 2018).

Line 20: source for the solution found in Eq (A3) to Eq (A2)?
We added a more explicit statement of the boundary conditions but feel reluctant to include a full derivation of the solution in the manuscript since space is limited. The curious reader may verify the validity of the solution by plugging Eq. (A3) into Eq. (A2).

Line 2: "Nevertheless…" Need citation or rationale for why Eq (A2) can be used to calculate $\delta^{15}N$
Thank you, an explanation is now included.

**Technical Corrections**

Line 14: "supresses" → suppresses
corrected

Line 2: "…unconsolidated snow…" I think of firn as consolidated snow.
corrected

Line 10: "processes which" → processes, which
sentence changed

Line 14: "high resolution" → high-resolution
corrected

Line 15: remove "the same amount of"
corrected

Line 1: "Last" → Lastly
corrected

Line 3: "gradients and induce" → gradients that induce
corrected

Line 17: "DZ but the effective" → DZ but effective
corrected

Line 7: change " becomes" to "simplifies to" (even though you already have "simplifies to" in the text just above)
corrected

Line 6: "Kawamura (unpublished)"- might as well leave this out since you have another citation
corrected

Line 4: "In the following we" → In the following discussion we
corrected

Line 9: "Isotopes ratios are higher" → Simulated isotope ratios are higher
corrected

Line 1: "the dispersivity" → dispersivity
corrected

Line 4: "estimate is within" → estimate of 2.8 m is within
corrected

Line 17: "…2013)as" → …2013) as
corrected

Line 13: "the isotopologues" → remove the "the"
corrected

Line 20: "at WAIS Divide" → at the WAIS Divide
corrected

Line 23: "75.9% almost" → 75.9%, almost
corrected

Line 22: remove the "This"
corrected

Line 23: remove "clearly"
corrected

Line 1: "these relationships" → these scaling relationships
corrected

Line 10: "Movment" → Movement
corrected

Line 11: "seperation" → separation
corrected

**Technical Corrections for Supplement**
Line 2: "q is a Nx1 vectors" → q is an Nx1 vector
corrected

Line 5: "off diagonals" → off-diagonals
corrected

Line 1: "at all depth" → at all depths
corrected

Fig S3: Does this figure indicate that there is a ~35 year delay in the response of the $\delta^{15}N$ profile due to the 3-day time step relative to the 3 day/5 ~14 hr time step?
~35 years is the adjustment time scale of the deep firn, i.e., the time it takes to bring the firn back into steady state if the firn conditions are changed slightly. For the plot, the model is initialized with the WAIS Divide

isotope profile simulated for 2006 and run for another 50 years with a 5x shorter time step. Note that in the previous version of the manuscript there was a small numerical problem with this initialization that has now been corrected. After this correction was implemented, the error now remains smaller than 0.1 per meg (previously 0.5 per meg) and converges after ~15 years to a level where further increase becomes undetectable compared to the interannual variability seen in the plot.

Line 3: "Eularian" → Eulerian
corrected

Fig S6: "Plot is shown at reduced grid resolution for clarity." Is this plot in reduced grid resolution or is it a subset of the domain at the original resolution (or both)?
The plot shows a selected region in the lock-in zone. A problem with the x-tick labels was corrected and the caption edited for clarity.

Note: there is a gap in the text partway down the page
corrected

Line 4: "firnf"?
Corrected

Line 6: is there a reference for equation S28?
A reference was added.

Line 1: "serval harmonic"?
Corrected to read "several harmonics"

Table S1: for the WAIS Divide the daily pressure change is 5hPa. Does this mean that for a 3-day time step the pressure change is 15hPa? Or is the 3-day pressure change no more than 15hPa and is quasi-randomly attenuated to match a red-shifted spectra? Or ?
The observed pressure variability at WAIS Divide is ~7hPa per day. (In the updated version of the model the amplitude attenuation was removed.) This forcing is implemented as random pressure changes every 3.5 days. The amplitude of this pressure variability must be increased by a factor of $\sqrt{\Delta t/day}$ to account for the longer time step of the model.

**Second, anonymous review**

**General comments**

The submitted manuscript analyses the influence of impermeable layer and barometric pumping (driven by surface pressure variability) on firn air transport using a 2D trace gas advection-diffusion-dispersion model accounting for discontinuous horizontal layers of reduced permeability. The simulated results are compared with field measurements from WAIS Divide and Law Dome DSSW20K and show good agreements.

**Specific comments**

The manuscript is well written and shows interesting results which are sufficient to be published. However, I would suggest to add some minor comments to make the manuscript clearer:

We thank the reviewer for the positive assessment of this study.

1. In the Figures, I would mention the time step you used for the plots. Are they consistent with the field measurements?

The time step for all model runs is 3.5 days unless otherwise noted owing to the high computational cost of running the 2D model. This is now mentioned in the main text. Sensitivity tests using a shorter time step are shown in the supplementary information.

2. In the 'supplementary information' you show a table with some simulation parameters, however some information is missing or is unclear, respectively. (e.g. simulation time, time range of the simulations, resolutions, etc.)

We agree that the table was not easily accessible. Item names in the table were edited for clarity (see also detailed comments below).

3. I would recommend to also show plots of the boundary conditions at the surface, like temperature, pressure, etc.

An exemplary time series of surface pressure fluctuations at WAIS Divide over the course of one year was added to the supplementary information.

We elect not to show a surface temperature history in the manuscript because we do not wish to suggest that our temperature histories are necessarily accurate reconstructions of past surface conditions at the site (shown here only for completeness). Our forcing histories are based on previously published temperature reconstructions (Dahl-Jensen et al., 1999; Van Ommen et al., 1999; Orsi et al., 2012) but adjusted slightly (in the case of WAIS Divide) to yield thermal gradients consistent with borehole temperature observations in our computationally simplified temperature model based on Alley and Koci, 1990 (Fig. RW2). Thermal diffusion only has a small effect on the simulated isotope profile below the surface mixed zone, therefore the details of the temperature forcing are mostly inconsequential.

Atmospheric $CO_2$ and $CH_4$ histories used to drive the model are presented in the supplementary information (Fig. S4).

[Figure]

**Figure RW2.** Surface temperature forcing of the model at WAIS Divide and Law Dome DSSW20K (offset by -8°C) without the seasonal cycle component. Our forcing generally agrees well with the more robust surface temperature reconstruction for WAIS Divide by Orsi et al. 2012.

**Detailed comments manuscript**

Page 3, Line 5 – 6: I would recommend to make 'z', 'T', and 'R' italic to make it consistent.
Font changed to italic

Page 7, Line 2 – 3: Have you checked the influence of airflow due to temperature changes, especially in the convective zone? Normally, in the top 10-15 meters you have a high variation in the temperature profile due to the changing temperature at the top. This variation in the temperature profile can cause an airflow.
We do not explicitly simulate temperature-driven convection in the surface mixed zone. Instead, convection and wind-driven mixing is represented by mass-independent "eddy" mixing. This follows common practice in the firn modeling community.

Page 11, Line 4: '…assumed to be temperature independent if the temperature sensitivity is unknown.' →
Where does this temperature sensitivity appear?
Omega is a function of temperature as described by eq. (11). The sentence was edited for clarity.

Page 11, Figure 7: There are two outliers in the observed CO2 concentrations at around 15 and 50 meters. Is there a reason for this?
These samples were likely contaminated with modern air before analysis and a comment on this issue has been added (see response to Review 1).

Page 11, Figure 7 caption: You refer to Fig. S9 to illustrate the differences in the CO2 and CH4 profiles between the 1D and the 2D model with or without barometric pumping which is not visible at the resolution of this Figure 7. However, Fig. S9 shows the same figure like Fig. 6 but only for Law Dome DSSW20K. Where can I find the Figure of the differences in the CO2 and CH4 profiles between the 1D and the 2D model with or without barometric pumping?
The Figure reference was changed to now correctly reference Figure S11 in the Supplementary Information (numbering of Figures changed).

Page 12, Line 10: See my comment to Page 11, Figure 7 caption.
corrected

Page 17, Line 13 – 14: What could be the reason that the simulated 86Kr excess is significant lower than the observed one?

In the discussion (section on the modified Péclet number), we explore the hypothesis that some unresolved subgrid-scale physics may be critical for obtaining larger $^{86}$Kr excess. Alternatively, a different representation of the surface mixed zone could lead to larger $^{86}$Kr excess simulated by the model than in the current setup. If the surface mixed zone was extended to great depth but the strength of mixing reduced to maintain agreement with the observed $CO_2$ profile, $^{86}$Kr excess below the surface mixed zone would increase in the model for example (the dark gray area in Figure 14 would expand). The importance of correctly representing the surface mixed zone will need to be further explored elsewhere.

Page 18, Line 6 – 7: 'Heavy, slow-diffusing isotopes approach gravitational equilibrium more slowly than lighter, faster-diffusing isotopes.' -> Maybe it is a silly question but is it not the opposite around? If the isotopes are heavy you have a faster settlement due to stronger gravitational force?

The reviewer is correct in that, heavier isotopes are indeed more gravitationally fractionated than lighter isotopes in firn because they experience a larger gravitational pull; the gravitational enrichment scales linearly with the mass difference between isotopes. However, isotopes approach gravitational equilibrium by diffusion (molecular diffusion drives the system towards chemical equilibrium. In firn air transport, chemical equilibrium must include the potential energy associated with the gravitational field of Earth) and the diffusivity of a gas typically decreases with it molecular weight. Therefore, heavy isotopes approach gravitational equilibrium more slowly.

Page 22, Line 24: 'However, our experiments fail to …' -> Which experiments? I cannot see in your manuscript that you did experiments, just simulations.

Wording changed to "numerical experiments".

Page 24, Line 3 and 9: I think you want to refer to Eq. (A3) instead to Eq. (A4).

The text was changed accordingly.

**Detailed comments supplementary information**

Page 2, Equation S5: '$q$' on the left-hand side of the equation is missing.

Corrected

Page 4, Line 15 – 20: Does it mean that snow accumulation is included?

Snow accumulation is constant in time and represented in the model as the vertical migration of the firn matrix including its gases and layers. Further details on the treatment of advection may be found in the SI.

Page 10, Line 4: How long does it take to run one simulation with time steps of around 3 days?

The runtime is approximately 2 days on 10 CPUs per simulation (with $\Delta t = 3.5$ days). A mention of this has been added.

Page 11, Section 'Thermal model': Can you show a figure of the temperature profile to get an impression of the boundary condition?

We added a borehole temperature profile with the simulated thermal gradient to the SI.

Page 11/12: The line break is wrong.

corrected

Page 12, Line 4: The line break is wrong and it should mean 'firn'

corrected

Page 12, Line 10: Please show a figure of the surface temperature histories.
See response to specific comments above.

Page 13, Table S1: Are the parameters the same for WAIS Divide and Law Dome DSSW20K if you only show one number, e.g. the 'Horizontal' or 'Vertical grid spacing'?
If only one number is show the value is the same for both sites. The table has been reformatted to clarify this.

Page 13, Table S1, Row 'Width': Does this mean there is a variation of the width? What are the exact values for WAIS Divide and Law Dome DSSW20K you used in the simulations?
The width of the model is different for each model to maintain a constant ratio between the thickness of annual layers and the width of the model. Exact values were added to the table.

Page 13, Table S1, Row 'Depth of first layer': Can you provide an exact value?
values added to table

Page 13, Table S1, Row 'Temperature': Did you use a fix temperature value at the top? No daily or seasonal variations?
The model includes anthropogenic warming and a seasonal temperature cycle (see SI Section 3). Diurnal temperature changes are neglected. The item in the table is the observed annual mean temperature at each site (item name corrected).

Page 13, Table S1, Row 'Surface Pressure: Did you use a fix pressure value at the top? No daily or seasonal variations?
Table S1 only provides the mean surface pressure and the daily variability of pressure at each location. A plot of surface pressure was added to the SI (Fig. S8) and the table item name edited for clarity.

Page 13, Table S1, Row 'Free air relative diffusivities to CO2': Can you show the value you cited from the Paper?
Because the model requires a considerable number of these free air diffusivity values, instead of reproducing them in the SI we now specifically reference the table in the original publication by Buizert et al. (2012).

Page 15, Line 5: '… using q-values.' -> please change it to '$q$-values'.
Manuscript corrected.

[revised manuscript text omitted]

**2.2  Barometric pumping**

The source term for barometric pumping, $\alpha_b$, is equal to the change in firn air density caused by surface pressure anomalies associated with passing storms (Fig. S8). Air compression or expansion demands a local convergence or divergence of flow that forces air to move in or out of the firn, assuming porosity remains constant.

[Figure]

**Figure S8.** One-year subsection of the surface pressure forcing driving barometric pumping at WAIS Divide. Observed daily pressure variability of ~7 mbar was rescaled by a factor of $\sqrt{\Delta t}$ to account for the longer time step used in the model.

[revised manuscript text omitted]

10 **Figure S9.** Observed borehole temperature and simulated temperature profile at WAIS Divide in January 2009 (Data from: Orsi et al., 2012).

**4 Summary tables**

**Table S1.** Overview of important model parameters

| Parameters | WAIS Divide | Law Dome DSSW20K |
|---|---|---|
| **Firn  depth (i.e., open** | 85 m | 65 m |
| **Model height ** | 80.92 m  | 52.74 m |
| **Model width** | 2.85 m (= 12x annual layer thickness) | 2.43 m |
| **Horizontal grid spacing** | 0.03 m | 0.03 m |
| **Vertical grid spacing** | 0.04 m | 0.04 m |
| **Depth of occurrence of first layer** | 56.65 m (= 70 % of firn | 36.94 m |
| **Simulation time range & time** | 1800–2006 in 3.5-day timesteps | 1800–1998.05 in 3.5-day timesteps |
| **Obs. annual mean temperature** | 243.15 K | 253.45 K |
| **Ice sheet  thickness H** | 3500 m | 1200 m |
| **Mean surface Pressure [1,2]** | 789 hPa | 850 hPa |
| **Surface pressure variability (1 $\sigma$)** | 7 hPa day[-1] | 11.2 hPa day[-1] |
| **Ice equiv. advection velocity[1,2]** | $6.9714 \times 10^{-9}$ m s[-1] | $5.1706 \times 10^{-9}$ m s[-1] |
| **Surface mixed zone eddy diffusion[3]** | $D_{e0} = 2.38 \times D_{m0}$ | $D_{e0} = 2.4 \times D_{m0}$ |
| $D_e = D_{e0} \exp\left(-\frac{z}{\tau}\right)$ | $\tau = 2.5$ m | $\tau = 3.5$ m |
| | range: 0 – 8 m + 2 m linear taper | range: 0 – 14 m + 2 m linear taper |

[1] WAIS Divide: WAIS Divide Project Members (2016)
[2] Law Dome: Etheridge et al. (1992)

[3] Eq. Kawamura et al. (2013)
[4]
[5]
[6]
[7]
[8]
[9]

5 **Table S2.** Overview of selected parameterizations in the model

| | |
|---|---|
| **Density of ice[1]** | $\rho_{ice} = 916.5 - 0.14438\,(T - 273.15) - 1.517 \cdot 10^{-4}(T - 273.15)^2$ kg |
| **Free air diffusivities relative to $CO_2$** | Table 4 & 5 in SI to Buizert et al. (2012) and references therein |
| **Dispersivity[2] (assumed isotropic)** | $\alpha\left(s_{op}\right) = 1.26 \cdot \exp(-25.7 s_{op})$ |
| **Total porosity** | $s_t = 1 - \dfrac{\rho_{firn}}{\rho_{ice}}$ |
| **Closed porosity[3]** | $s_{cl} = 0.37 \cdot s_t \left(\dfrac{s_t}{1 - \dfrac{831.2}{\rho_{ice}}}\right)^{-7.6}$ |
| **Firn density fit** $\leq z_{crit1}$ | $\rho_{firn} = a_0 + a_1 z + a_2 \cdot \exp[a_3 \cdot (z_{crit1} - z)]$ kg m[-3] |
| $z_{crit1} - z_{crit2}$ | $\rho_{firn} = b_0 + b_1 z + b_2 z^2$ kg m[-3] |
| $\geq z_{crit2}$ | $\rho_{firn} = \rho_{ice} - \left(\rho_{ice} - \rho(z_{crit2})\right) \cdot \exp\left[-\frac{z - z_{crit2}}{\rho_{ice} - \rho(z_{crit2})}(b_1 + 2 \cdot b_2 z_{crit2})\right]$ kg m[-3] |

[revised manuscript text omitted]